# Histone deacetylase 8 interacts with the GTPase SmRho1 in *Schistosoma mansoni*

**Lucile Pagliazzo**[1], **Stéphanie Caby**[1], **Julien Lancelot**[1], **Sophie Salomé-Desnoulez**[2], **Jean-Michel Saliou**[2], **Tino Heimburg**[3], **Thierry Chassat**[4], **Katia Cailliau**[5], **Wolfgang Sippl**[3], **Jérôme Vicogne**[1]*, **Raymond J. Pierce**[1]*

**1** Univ. Lille, CNRS, Inserm, CHU Lille, Institut Pasteur de Lille, - Centre d'Infection et d'Immunité de Lille, Lille, France, **2** Univ. Lille, CNRS, Inserm, CHU Lille, Institut Pasteur de Lille, Lille, France, **3** Institute of Pharmacy, Martin-Luther University of Halle-Wittenberg, Halle/Saale, Germany, **4** Institut Pasteur de Lille - PLEHTA (Plateforme d'expérimentation et de Haute Technologie Animale), Lille, France, **5** Univ. Lille, CNRS, UMR 8576-UGSF-Unité de Glycobiologie Structurale et Fonctionnelle, Lille, France

* jerome.vicogne@ibl.cnrs.fr (JV); raymond.pierce@pasteur-lille.fr (RJP)

**Data Availability Statement:** All relevant data are within the manuscript and its Supporting information files.

## Abstract

### Background

*Schistosoma mansoni* histone deacetylase 8 (SmHDAC8) has elicited considerable interest as a target for drug discovery. Invalidation of its transcripts by RNAi leads to impaired survival of the worms in infected mice and its inhibition causes cell apoptosis and death. To determine why it is a promising therapeutic target the study of the currently unknown cellular signaling pathways involving this enzyme is essential. Protein partners of SmHDAC8 were previously identified by yeast two-hybrid (Y2H) cDNA library screening and by mass spectrometry (MS) analysis. Among these partners we characterized SmRho1, the schistosome orthologue of human RhoA GTPase, which is involved in the regulation of the cytoskeleton. In this work, we validated the interaction between SmHDAC8 and SmRho1 and explored the role of the lysine deacetylase in cytoskeletal regulation.

### Methodology/principal findings

We characterized two isoforms of SmRho1, SmRho1.1 and SmRho1.2. Co- immunoprecipitation (Co-IP)/Mass Spectrometry (MS) analysis identified SmRho1 partner proteins and we used two heterologous expression systems (Y2H assay and *Xenopus laevis* oocytes) to study interactions between SmHDAC8 and SmRho1 isoforms.

To confirm SmHDAC8 and SmRho1 interaction in adult worms and schistosomula, we performed Co-IP experiments and additionally demonstrated SmRho1 acetylation using a Nano LC-MS/MS approach. A major impact of SmHDAC8 in cytoskeleton organization was documented by treating adult worms and schistosomula with a selective SmHDAC8 inhibitor or using RNAi followed by confocal microscopy.

### Conclusions/significance

Our results suggest that SmHDAC8 is involved in cytoskeleton organization *via* its interaction with the SmRho1.1 isoform. The SmRho1.2 isoform failed to interact with SmHDAC8,

**Funding:** This work and the the authors LP, SC, JL, TH, WS, JV and RJP have been supported by funding from the European Union's Seventh Framework Programme for research, technological development and demonstration under grant agreement no. 602080 (AParaDDisE). Authors LP, SC, JL, S S-D, J-MS, TH, KC, JV and RJP were supported by institutional funds from the CNRS UMR 9017, the Institut Pasteur de Lille and Lille University. The funders had no role in study design, data collection and analysis, decision to publish or preparation of the manuscript.

**Competing interests:** The authors have declared that no competing interests exist.

but did specifically interact with SmDia suggesting the existence of two distinct signaling pathways regulating *S. mansoni* cytoskeleton organization *via* the two SmRho1 isoforms. A specific interaction between SmHDAC8 and the C-terminal moiety of SmRho1.1 was demonstrated, and we showed that SmRho1 is acetylated on K136. SmHDAC8 inhibition or knockdown using RNAi caused extensive disruption of schistosomula actin cytoskeleton.

## Author summary

*Schistosoma mansoni* is the major parasitic platyhelminth species causing intestinal schistosomiasis. Currently one drug, praziquantel, is the treatment of choice but its use in mass treatment programs means that the development of resistance is likely and renders imperative the development of new therapeutic agents. As new potential targets we have focused on lysine deacetylases, and in particular *S. mansoni* histone deacetylase 8 (SmHDAC8). Previous studies showed that reduction in the level of transcripts of SmHDAC8 by RNAi led to the impaired survival of the worms after the infection of mice. The analysis of the 3D structure of SmHDAC8 by X-ray crystallography showed that the catalytic domain structure diverges significantly from that of human HDAC8 and this was exploited to develop novel potential anti-schistosomal drugs. The biological roles of SmHDAC8 are unknown. For this reason, we previously characterized its protein binding partners and identified the schistosome orthologue of the human RhoA GTPase, suggesting the involvement of SmHDAC8 in the modulation of cytoskeleton organization. Here we investigated the interaction between SmHDAC8 and SmRho1 and identified two SmRho1 isoforms (SmRho1.1 and SmRho1.2). Our study showed that SmHDAC8 is involved in schistosome cytoskeleton organization.

## Introduction

Schistosomiasis is a Neglected Tropical Disease and represents the second most important human parasitic disease after malaria [1, 2]. It is caused by flatworm parasites of the genus *Schistosoma* and more than 200 million people are infected in 76 countries [3]. Treatment and control of the disease depends on the only available drug, Praziquantel (PZQ). PZQ is effective specifically on adult worms and against the three major species of schistosomes infecting humans (*S. mansoni*, *S. haematobium and S. japonicum* [4]). However, its massive administration in endemic areas, in monotherapy, has promoted the emergence of PZQ-tolerant parasites, rendering more likely the risk of resistance [5–8], and the development of new therapeutic agents imperative.

Lysine deacetylases (KDACs) also called Histone deacetylases (HDACs) form a family of enzymes that are conserved in metazoans [9]. They are attractive therapeutic targets because they are involved in the regulation of gene transcription and are already actively studied as drug targets in other pathologies, particularly cancer [10]. Our previous studies identified and characterized three class I HDACs in *S. mansoni*: HDAC 1, 3 and 8 [11] and we have shown that Trichostatin A (TSA), a pan-inhibitor of HDACs, induces hyperacetylation of histones, deregulates gene expression and causes the death of schistosome larvae and adult worms in culture [12]. Moreover, reduction of *SmHDAC8* transcripts by RNAi led to the impaired survival of the worms after the infection of mice, showing that it is a valid therapeutic target [13]. The analysis of the 3D structure of SmHDAC8 by X-ray crystallography [13] showed that the

catalytic domain structure diverged significantly from that of the human HDAC8 and this was exploited to identify selective inhibitors that induce apoptosis and death of worms and are thus lead compounds for the development of novel anti-schistosomal drugs [14]. However, the precise biological roles of schistosomal HDAC8 were unknown and in order to determine why SmHDAC8 knockdown or inhibition causes cell apoptosis and worm death, it was essential to study the cellular signaling pathways involving SmHDAC8. Potential binding partners of SmHDAC8 have been characterized by screening a yeast two-hybrid cDNA library and mass spectrometry analysis [15]. Among the potential partners identified, the schistosome orthologue of the human GTPase RhoA, SmRho1, indicated a potential role for SmHDAC8 in cytoskeleton organization [15].

Rho GTPases (Ras homologous) belong to the superfamily of small Ras (Rat Sarcoma) monomeric G proteins that are extremely conserved in eukaryotes [16]. They are able to create a switch between an active GTP-bound conformation and an inactive GDP-bound conformation. The "ON/OFF" activity of Rho GTPase is controlled by various regulatory proteins: the guanine nucleotide exchange factors (GEFs) induce the exchange of GDP for GTP; the GTPase-activating proteins (GAPs) promote the hydrolysis of GTP to GDP and the GDP dissociation inhibitors (GDIs) inhibit the dissociation of GDP from the GTPase [17]. Activation of Rho GTPase by GEFs transduces signals to various effector molecules, while remaining in the GTP-bound form, hence regulating various cell functions through reorganization of the actin cytoskeleton such as the formation of stress fibers and focal adhesions [18]. In *S. mansoni*, SmRho1 was identified and presents 71–75% identity to human RhoA, B, and C GTPases. It was also able to complement a Rho1 null mutation in the budding yeast *Saccharomyces cerevisiae* [19] and could play a role in actin cytoskeleton regulation in the gonads of adult worms [20]. Although HDAC8 is known to interact with specific cytoskeleton components [21], a direct link between HDAC8 and a Rho GTPase has never been shown in humans. Moreover, RhoA was not identified by Olson *et al.* as a partner of human HDAC8 [22]. Interestingly, RhoA does not seem to be acetylated in humans, although its orthologue was found to be acetylated in *S. japonicum* [23].

In the present study, two closely related isoforms of SmRho1, which we named SmRho1.1 and SmRho1.2, were characterized at the molecular and functional levels. Based on these data, we focused on the interaction between SmRho1 isoforms and SmHDAC8. Studies in the parasite showed that SmHDAC8 interacts with SmRho1. Using mass spectrometry analysis, we identified an acetylated lysine on SmRho1 in adult worms. Co-expression of SmRho1 isoforms and SmHDAC8 in *X. laevis* oocytes and in yeast shows a specific interaction between SmRho1.1 and SmHDAC8 that implicates the SmRho1.1 C-terminal domain. In the same way, we showed that SmRho1.2 co-immunoprecipitates with a Diaphanous homolog SmDia [24] and not with SmHDAC8. Finally, the use of selective inhibitors and RNAi followed by confocal microscopy revealed that SmHDAC8 is involved in the regulation of the actin cytoskeleton organization in adult worms and in schistosomula. Hence, here we demonstrate, for the first time, the role of SmHDAC8 in modulating the organization of the schistosome cytoskeleton, possibly *via* the SmRho GTPase signaling pathway.

## Methods

### Ethical statement

All animal experimentation was conducted in accordance with the European Convention for the Protection of Vertebrate Animals used for Experimental and other Scientific Purposes (ETS No 123, revised Appendix A) and was approved by the committee for ethics in animal experimentation of the Nord-Pas de Calais region (Authorization No. APAFIS#82892016122015127050v3) and the Pasteur Institute of Lille (Agreement No. B59350009). Experiments on *X. laevis* were

performed according to the European Community Council guidelines (86/609/EEC). The protocols were approved by the institutional local "Comité d'Ethique et d'Expérimentation Animale, Région Haut de France, F59-00913".

## Parasite material

The NMRI strain of *S. mansoni* is maintained in the laboratory using the intermediate snail host *Biomphalaria glabrata* and the definitive golden hamster host *Mesocricetus auratus*. *S. mansoni* adult worms were obtained by hepatic portal perfusion of hamsters infected six weeks before. Cercariae were released from infected snails, harvested on ice as described in [25] and schistosomula were obtained *in vitro* by mechanical transformation [25].

## Frog and oocytes handling

*X. laevis* females, obtained from the CRB-University of Rennes (France), were anesthetized with tricaine methane sulfonate (MS222, Sandoz) at 1 gL$^{-1}$ for 45 min. After surgical removal of ovaries, stage VI oocytes were harvested by using a 1 h collagenase A treatment (1 mg mL$^{1}$, Boehringer Mannheim) for 45 minutes followed by manual dissociation in ND96 medium (96 mM NaCl, 2 mM KCl, 1 mM MgCl$_2$, 1.8 mM CaCl2, 5 mM HEPES, adjusted to pH 7.5 with NaOH). Oocytes were kept at 19°C for 2h.

## Molecular cloning of SmRho1 isoforms

Total RNA from adult worms was isolated using the RiboPure RNA Purification Kit (Thermofisher Scientific). The cDNA was prepared using the GeneRacer kit with SuperScript III reverse transcriptase (Invitrogen) following the manufacturer's instructions. The 5 'and 3' ends of *SmRho1.1* and *SmRho1.2* were determined by RACE PCR using the primers listed in S1 Table (Rho GTPase 5'1 / 5'2 and 3'1 / 3'2) generated from the cDNA sequences obtained by Y2H screening (15) and amplified products were subcloned into the vector pCR4-TOPO (Thermofisher Scientific) and sequenced (Eurofins Genomics). The full length *SmRho1* isoform sequences were amplified using FLRho1 5' and 3' primers and inserted into the pCR4-TOPO vector. A further PCR experiment was then carried out, using primers containing the *Nde*I and *Bam*HI restriction sites respectively and the obtained fragment was again inserted into the pCR4-TOPO vector.

## Phylogenetic analysis and protein modeling

Phylogenetic analysis of eukaryotic Rho GTPases was performed using protein sequences from vertebrates, helminths (nematodes, cestodes, turbellariates), insects, molluscs and yeasts (S2 Table). The sequences were aligned using the BioEdit program using the ClustalW method [26]. The phylogenetic tree was generated by the MEGAX software under the JTT + I + G model with 1 000 bootstraps [27] and the maximum likelihood method. The modeling of the two isoforms of SmRho1 was performed using the I-Packer server (https://zhanglab.ccmb.med.umich.edu/I-TASSER/) and compared to the structure of human RhoA [28]. The characteristic domains and the different residues between SmRho1.1, SmRho1.2 and *Homo Sapiens* RhoA (HsRhoA) were highlighted using PYMOL software [29].

## Expression and purification of the SmRho1.1 and SmRho1.2 recombinant proteins

SmRho1.1-pGADT7 and SmRho1.2-pGADT7 constructs were cut by *Nde*I and *Bam*HI. Sequences were inserted in frame into the pnEA-tH plasmid [13] (a kind gift from Dr. M.

Marek and Dr. C. Romier, IGBMC, Strasbourg, France, which codes for a polyhistidine tag in the N-terminal position followed by a thrombin site), using T4 DNA Ligase (Invitrogen).

Overexpression was carried out in *E. coli* BL21(DE3) cells in Luria Broth (LB) medium. Induction was done at 37˚C by adding isopropyl-1-thio-β-D-galactopyranoside (IPTG, Euromedex) to a final concentration of 500 mM. Harvested bacteria were resuspended in lysis buffer (400 mM KCl, 10 mM Tris-HCl pH 8.0) and protease inhibitors (20 µM leupeptin (Sigma), 2 µg mL$^{-1}$ aprotinin (Sigma), 200 µM phenylmethylsulfonyl fluoride (PMSF, Sigma) and sonicated at 4˚C, 30 times for 30 s (maximum power, Bioruptorplus, Diagenode). The lysate was clarified by ultracentrifugation (41 000 x G) for 1 h at 4˚C. The supernatant was loaded onto Nickel affinity resin (Clontech) pre-equilibrated with the lysis buffer and His tagged SmRho1 proteins were released from the nickel resin by imidazol treatment. Protein concentrations were measured using the Pierce BCA Protein Assay Kit (Thermo Fisher Scientific).

### Production of anti-SmRho1 antibodies

Purified recombinant SmRho1.1 was used to generate mouse polyclonal antiserum. BALB/c mice were injected i.p. with 50 µg of SmRho1.1 and alum adjuvant in a total volume of 500 µL, three times at two-week intervals. The mice were bled two weeks after the final injection. The activity of the mouse antiserum was controlled on *S. mansoni* protein extract at all parasitic stages and on SmRho1 recombinant proteins by Western Blot (WB) (S2 Fig). The antiserum recognized both recombinant SmRho1.1 and SmRho1.2 as expected given the high level of identity between the two protein sequences (S2B Fig). WB of proteins from different schistosome life-cycle stages (S2C Fig) shows the detection of a major band at the expected molecular weight, but also some additional bands. Low molecular weight bands may correspond to other SmRho family members, notably SmRho2, due to their relatively high level of identity to the SmRho1 isoforms. Nevertheless, the antiserum recognized the SmRho1 isoforms in schistosome extracts, corresponding to the major band at 22 kDa in WB (S2 Fig) and demonstrated by the results of CoIP/MS. In the latter experiments SmRho1-derived peptides were a major element among the detected proteins (S1 Fig and S3 Table).

In order to further establish the specificity of the anti-SmRho1 antibodies, we performed an immunodepletion assay (S2D Fig). Briefly, 100 µg of SmRho1.1 recombinant protein were incubated with 10 µL of SmRho1 antiserum for 2h at room temperature on a rotating wheel. After TRIzol extraction (Invitrogen), 20 µg of protein extracts from adult worms and schistosomula were separated on a 10% (v/v) SDS–polyacrylamide gel and blotted on to a nitrocellulose membrane. Blots were developed with a mouse polyclonal anti-SmRho1 antibody (1/1000) or mouse anti-SmRho1 depleted serum and peroxidase coupled anti-mouse secondary antibody (1/50 000; Invitrogen). An antibody against human RhoA (Rabbit anti-HA protein tech group cat: PTG10749-1-Au, 1/1000) was used as a negative control.

Detection was carried out by chemiluminescence using SuperSignal West Dura Extended Duration Substrate (Thermo Scientific) and ImageQuant LAS 4000 imager (GE Healthcare). Results showed that only the major band at 22 kDa was depleted in the assay, supporting the recognition of SmRho1 by the mouse antiserum.

### CoIP for Nano-LC–MS/MS analysis

For mass spectrometry analysis, two independent experiments were performed. Adult worms (100 couples) were suspended in 500 µL of lysis buffer (20 mM Tris-HCl pH 7.4, 50 mM NaCl, 5 mM EDTA, 1% Triton and protease inhibitors (20 µM leupeptin (Sigma), 2 µg mL$^{-1}$ aprotinin (Sigma), 200 µM phenylmethylsulfonyl fluoride (Sigma), crushed with a Dounce homogenizer

and sonicated ten times for 30 s (maximum power, Bioruptorplus, Diagenode). After centrifugation at 1000 x G for 10 min at 4°C, immunoprecipitation of SmRho1 was performed using the Pierce Crosslink Immunoprecipitation Kit (Thermo Scientific) according to the manufacturer's instructions. Briefly, the protein lysate (500 µL) was pre-cleared by incubation with 20 µL of IgG from rat serum crosslinked to protein-L Agarose beads (5 mg of beads, Thermo Scientific) for 2h at 4°C on a rotator. Then, pre-cleared lysate was collected after centrifugation, at 1000 x G for 1 min at 4°C, and incubated overnight at 4°C on a rotator, with 10 µL of anti-SmRho1 antibodies or 10 µL of IgG from mouse serum as a control, bound to protein-L Agarose beads (5 mg of beads, Thermo Scientific).

## Mass-spectrometry proteomic analysis

Protein samples were denatured at 100° C in 5% SDS, 5% β-mercaptoethanol, 1 mM EDTA, 10% glycerol, and 10 mM Tris pH 8 buffer for 3 min, and subsequently fractionated on a 10% acrylamide SDS-PAGE gel. The gel was stained briefly with Coomassie Blue. Five bands, containing the whole sample, were cut out. Digestion of proteins in the gel slices was performed as previously described [30]. Separation of the protein digests was carried out using an UltiMate 3000 RSLCnano System (Thermo Fisher Scientific). Peptides were automatically fractionated onto a commercial C18 reversed phase column (75 µm × 150 mm, 2 µm particle, PepMap100 RSLC column, Thermo Fisher Scientific, temperature 35°C). Trapping was performed during 4 min at 5 µL min$^{-1}$, with solvent A (98% $H_2O$, 2% ACN (acetonitrile) and 0.1% FA (Formic Acid)). Elution was carried out using two solvents, A (0.1% FA in water) and B (0.1% FA in ACN) at a flow rate of 0,3 mL/min. Gradient separation was 3 min at 5% B, 37 min from 5% B to 30% B, 5 min to 80% B, and maintained for 5 min. The column was equilibrated for 10 min with 5% buffer B prior to the next sample analysis. Peptides eluted from the C18 column were analyzed by Q-Exactive instruments (Thermo Fisher Scientific) using an electrospray voltage of 1.9 kV, and a capillary temperature of 275°C. Full MS scans were acquired in the Orbitrap mass analyzer over the m/z 300–1200 range with a resolution of 35 000 (m/z 200) and a target value of 5.00E + 05. The ten most intense peaks with charge state between 2 and 4 were fragmented in the HCD collision cell with normalized collision energy of 27%, and tandem mass spectra were acquired in the Orbitrap mass analyzer with resolution 17,500 at m/z 200 and a target value of 1.00E+05. The ion selection threshold was 5.0E+04 counts, and the maximum allowed ion accumulation times were 250 ms for full MS scans and 100 ms for tandem mass spectrum. Dynamic exclusion was set to 30 s. Raw data collected during nanoLC MS/MS analyses were processed and converted into *.mgf peak list format with Proteome Discoverer 1.4 (Thermo Fisher Scientific). MS/MS data were interpreted using search engine Mascot (version 2.4.0, Matrix Science, London, UK) installed on a local server. Searches were performed with a tolerance on mass measurement of 0.2 Da for precursor and 0.2 Da for fragment ions, against a composite target decoy database (25 970 total entries) built with the *S. mansoni* Uniprot database (taxonomy id 6183, 12 861 entries) fused with the sequences of recombinant trypsin and a list of classical contaminants (124 entries). Up to one trypsin missed cleavage was allowed. For each sample, peptides were filtered out according to the cut-off set for protein hits with one or more peptides longer than nine residues, an ion score >30, an identity score >6, leading to a protein false positive rate of 0.8%.

## IP and WB analyses of proteins expressed in adult *S. mansoni*

Adult worms (100 couples) and schistosomula (1000 parasites) were suspended in lysis buffer (20 mM Tris-HCl pH 7.4, 50 mM NaCl, 5 mM EDTA, 1% Triton and protease inhibitors: (20 µM leupeptin (Sigma), 2µg mL$^{-1}$ aprotinin (Sigma), 200 µM PMSF (Sigma) crushed with a

Dounce homogenizer and sonicated ten times for 30 s (maximum power, Bioruptoplus, Diagenode). After centrifugation, at 10 000 x G for 10 min at 4°C, immunoprecipitation of SmHDAC8 and SmRho1 was performed using the Pierce Crosslink Immunoprecipitation Kit (Thermo Scientific) according to the manufacturer's instructions. Briefly, protein lysate (500 μL) was incubated with 10 μL of mouse polyclonal anti-SmHDAC8 or mouse polyclonal antiSmRho1 crosslinked to protein-L Agarose beads (Thermo Scientific) overnight at 4°C on a rotating wheel. As a negative control, a Co-IP with a mouse IgG antibody alone was performed. Proteins were separated on a 10% (v/v) SDS–polyacrylamide gel and blotted on to a nitrocellulose membrane. Blots were developed with a mouse polyclonal anti-SmHDAC8 antibody (1/1000) or mouse polyclonal anti-SmRho1 antibody (1/1000) and peroxidase coupled anti-mouse secondary antibody (1/50 000; Invitrogen). Detection was carried out by chemiluminescence using SuperSignal West Dura Extended Duration Substrate (Thermo Scientific) and ImageQuant LAS 4000 imager (GE Healthcare).

## Plasmid constructs

The sequence encoding SmHDAC8 was inserted in frame into the pGBKT7-BDB expression vector (Clontech) using the T4 DNA Ligase (Invitrogen) to generate the SmDAC8-pGBKT7 construct as previously described (15). SmRho1.1-pCR4-TOPO and SmRho1.2-pCR4-TOPO were cut by *Bam*HI and *Nde*I and the sequences were inserted in frame into the pGADT7-AD expression vector (Clontech) using the T4 DNA Ligase (Invitrogen) (SmRho1.1-pGADT7 and SmRho1.2-pGADT7).

   Site-directed mutagenesis of SmRho1 mutant constructs was performed using the Isis DNA polymerase (MP Biomedicals). The SmRho1.1-pGADT7 construct was used as a template for the production of the SmRho1.1 E33K mutant in which the glutamic acid at position 33 is replaced by a lysine residue, using as primers SmRho1.1 E33K 5'/ SmRho1.1 E33K 3'. Similarly, the SmRho1.2 K33E construct was obtained using as primers SmRho1.2 K33E 5'/ SmRho1.2 K33E 3' and SmRho1.2-pGADT7 as template. SmRho1.1–88 aa and SmRho1.1143 aa fragments were obtained using respectively SmRho1.1–88 aa 5'/SmRho1.1–88 aa 3' and SmRho11-143 aa 5'/SmRho1.1–143 aa 3' as primers to generate a stop codon. For SmRho1.1 EM and SmRho1.1 EMNN mutants, the glutamine Q147 and the valine V148 of SmRho1.1 were substituted by a glutamic acid and a methionine (SmRho1.1 EM) and then the lysine K151 and the serine S153 by two asparagine residues (SmRho1.1 EMNN). Similarly, SmRho1.2 QV and SmRho1.2 QVKS mutants were produced by site-directed mutagenesis using the SmRho1.2 construct. First, the glutamic acid E147 and the methionine M148 were substituted by a glutamine and a valine and then, the two asparagines N151 and N153 were replaced respectively by a lysine and a serine.

## Yeast two hybrid assay

The *S. cerevisiae* Y187 strain was transformed with the SmHDAC8-pGBKT7 bait construct and mated overnight with the AH109 strain transformed with the SmRho1 construct. After incubation, diploid yeasts were plated on selective medium lacking leucine and tryptophan and then on selective medium lacking adenine, histidine, leucine and tryptophan and the plates were incubated at 30°C.

## *In vitro* synthesis of cRNAs

cRNAs encoding SmHDAC8, SmRho1 (SmRho1.1, SmRho1.2, mutants SmRho1.1 E333K, SmRho1.2 K33E, 1–88 aa and 1–143 aa, SmRho1.1 EM and SmRho1.1 EMNN, SmRho1.2 QV and SmRho1.2 QVKS) and SmDia-RDB (a kind gift from Prof. C. Grevelding, Institute of

Parasitology, Justus-Liebig-University, Giessen, Germany) were synthesized using the T7 mMessage mMachine kit (Ambion, USA). The SmHDAC8-pGBKT7, SmRho1-pGADT7 (SmRho1.1, SmRho1.2) and SmDia-RBD-pGBKT7 constructs were linearized using the restriction enzyme *Not*I. The SmRho1 mutant constructs were linearized using the restriction enzyme *Hin*dIII. cRNAs were precipitated by 2.5 M LiCl, washed in 70% ethanol, suspended in 20 mL diethylpyrocarbonate (DEPC)-treated water, and quantified by spectrophotometry. cRNAs were analyzed in a denaturing agarose gel. Gel staining with 10 mg mL$^{-1}$ ethidium bromide confirmed correct sizes and of absence of abortive transcripts. cRNA preparations (1 mg mL$^{-1}$) were microinjected into *X. laevis* oocytes (stage VI) as follows. Each oocyte was injected with 60 nL of cRNA in the equatorial region and incubated at 19˚C in ND96 medium supplemented with 50 mg mL$^{-1}$ streptomycin/penicillin, 225 mg mL$^{-1}$ sodium pyruvate, 30 mg mL$^{-1}$ trypsin inhibitor) for 18 h.

## Co-immunoprecipitation (CoIP) and WB analysis of proteins expressed in *X. laevis* oocytes

Immunoprecipitation of SmHDAC8, SmDia-RBD and SmRho1 (isoforms and mutant constructs) proteins expressed in oocytes was performed using HA and Myc tags respectively. 15h after cRNA injection in the equatorial region, oocytes were lysed in buffer (50 mM HEPES pH 7.4, 500 mM NaCl, 0.05% SDS, 0.5% Triton X100, 5 mM MgCl2, 1 mg mL$^{-1}$ bovine serum albumin, 10 mg mL$^{-1}$ leupeptin, 10 mg mL$^{-1}$ aprotinin, 10 mg mL$^{-1}$ soybean trypsin inhibitor, 10 mg mL$^{-1}$ benzamidine, 1 mM sodium vanadate) and centrifuged at 4˚C for 15 min at 10 000 x G. Supernatants were incubated with anti-Myc (1/100; Invitrogen) and anti-HA (1/100 Invitrogen) antibodies for 4 h at 4˚C. Protein A-Sepharose beads (5 mg, Amersham Biosciences) were added for 1 h at 4˚C. Immune complexes were collected by centrifugation, rinsed three times, resuspended in Laemmli sample buffer, and subjected to a 10% SDS-PAGE. Immune complexes were analyzed by Western Blotting using anti-Myc (1/50 000, Sigma Aldrich) or anti-HA (1/10 000, Invitrogen) antibodies and the advanced ECL detection system (Amersham Biosciences).

## RNAi-mediated knockdown of *SmHDAC8* and *SmRho1*

The SmHDAC8 specific PCR primers containing the T7 promoter-tail amplified ~500 bp fragments (S1 Table). Similarly, the SmRho1 specific PCR primers containing the T7 promoter tail amplified a ~500 bp fragment (S1 Table). A luciferase nonspecific ~500 bp control was used (S1 Table). Double-stranded RNAs (dsRNAs) were synthesized *in vitro* from adult worm cDNA using the Megascript RNAi kit (Ambion) according to the manufacturer's instructions and concentrations were determined spectrophotometrically (NanoVue Plus, GE Healthcare). dsRNAs were also analyzed by 1% agarose electrophoresis to check the integrity and annealing of the dsRNA transcripts. To deliver the dsRNA, 10 adult couples/group in 100 μL M199 medium containing 25 μg dsRNA, were electroporated in a 4 mm cuvette by applying a square wave with a single 20 ms impulse, at 125 V and at room temperature [31]. Adult worms were then transferred to 4 mL complete M199. After two days in culture, 2 mL of medium was removed and 2 mL of fresh complete M199 culture medium was added. Gene knockdown was monitored by qRT-PCR 5 days after dsRNA treatment. For RNAi experiments on schistosomula, dsRNA delivery was performed using the soaking method. 10 μg of dsRNA was added to 2000 parasites in 4 ml complete M199 medium and after two days in culture, gene knockdown was monitored by qRT-PCR. Microscopic examination of RNAi-treated worms was carried out exactly as described below.

## Quantitative RT-PCR

Complementary DNAs were obtained by reverse transcription of total RNA using the NucleoSpin RNA/Protein kit (Macherey-Nagel) and used as templates for PCR amplification using Brilliant III Ultra-Fast QPCR Master Mix (Agilent) and QuantStudio 3 Real-Time PCR System (Applied Biosystems). Primers specific for *S. mansoni HDAC8*, *Rho1.1* and *Rho1.2* were designed by the Primer BLAST tool (NCBI) and used for amplication in duplicate assays (S1 Table). Measurements of real time PCR efficiency for each primer pair allowed the ratios of expression to be calculated using the $2^{-DDCt}$ ratio with *S. mansoni PSMB7* (Smp_073410) as the reference transcript [32]. Each sample was run in duplicate and the experiment was performed once. The results were expressed as means ± S.D.

## Phenotypic analysis by confocal laser scanning microscopy

Adult worms (10 couples) were treated with sub-lethal doses of inhibitors; for 16 h with TH65 [33], Trichostatin A (TSA) or Rho Inhibitor I at 50 μM, 10 μM and 160 μM respectively. Similarly, schistosomula (2000 parasites) were treated with TH65 (10 μM), TSA (3 μM) dissolved in Dimethyl sulfoxide (DMSO) or Rho Inhibitor I (80 μM) dissolved in $H_2O$ for 16h. Parasites treated with corresponding amounts of DMSO or $H_2O$ were used as negative controls.

After treatment, parasites were fixed in 8% PFA (paraformaldehyde)—CB buffer solution Cytoskeletal Buffer: 10 mM Hepes, 150 mM NaCl, 5 mM EGTA, 5 mM glucose, 5 mM MgCl2, pH 6,1), for 1h at room temperature. The fixed parasites were then incubated with CB buffer containing 0.05% saponin, NH4Cl (50 mM) and phalloidin (Alexa Fluor 488, Thermo Fischer Scientific, at 1/1000 dilution) overnight at room temperature to stain actin filaments. Three washes were performed in CB buffer and DAPI (1/1000 dilution, Thermo Fischer Scientific) was added during the third wash for 10 min at room temperature. Mowiol (Calbiochem) was used as mounting solution. Samples were pictured with a Zeiss LSM 880 confocal line scanning microscope (Zeiss microscopy GmbH, Germany) using an Airyscan detector and 63x oil immersion lens to obtain high-resolution images (voxel 0.35x0.35x0.30 $\mu m^3$). Images were processed using Zen software (Version2, Zeiss, France) for Airyscan processing. The visualization of the samples in depth were obtained by maximum intensity projection on an adapted selection of Z planes of the Z-stack and using orthogonal views created with FIJI (version: 2.1.1/1.53c).

## Results

### Identification and characterization of the two SmRho1 isoforms

Among the Rho family of small proteins (~21 kDa) the R̲as h̲omolog family, member A̲, (RhoA), R̲as-related C̲3 botulinum toxin substrate 1 (Rac1) and C̲ell D̲ivision C̲ontrol protein 42 homolog (Cdc42) are the most studied members. Santos and coworkers [34] characterized Rho1 in *S. mansoni*, showing that it is related to the vertebrate RhoA, B, C subfamily and that SmRho1 complemented an *S. cerevisiae* Rho1 null strain. The SmRho family was further characterized by Vermeire and coworkers [19] and they identified *S. mansoni* orthologs of RhoA, CDC42 and Rac1. Interestingly, using the yeast two-hybrid methodology and mass spectrometry analysis, we recently identified the *S. mansoni* Ras homolog 1 (SmRho1), as a SmHDAC8 partner protein [15]. Moreover, after a more detailed reconstruction of the *SmRho1* transcript using RT-PCR, we discovered that two distinct genes encode Rho1 isoforms: *Rho1.1* and *1.2*. These two genes code for two proteins of 193 residues organized into several domains, as for all Rho GTPases (Fig 1A). ClustalW alignment shows 96.37% identity between the two encoded proteins that differ only in 7 aa, and they share respectively 72% and 73% identity with the *Homo sapiens* RhoA (HsRhoA). The Rho GTPases contain five GTPase domains named "G boxes"

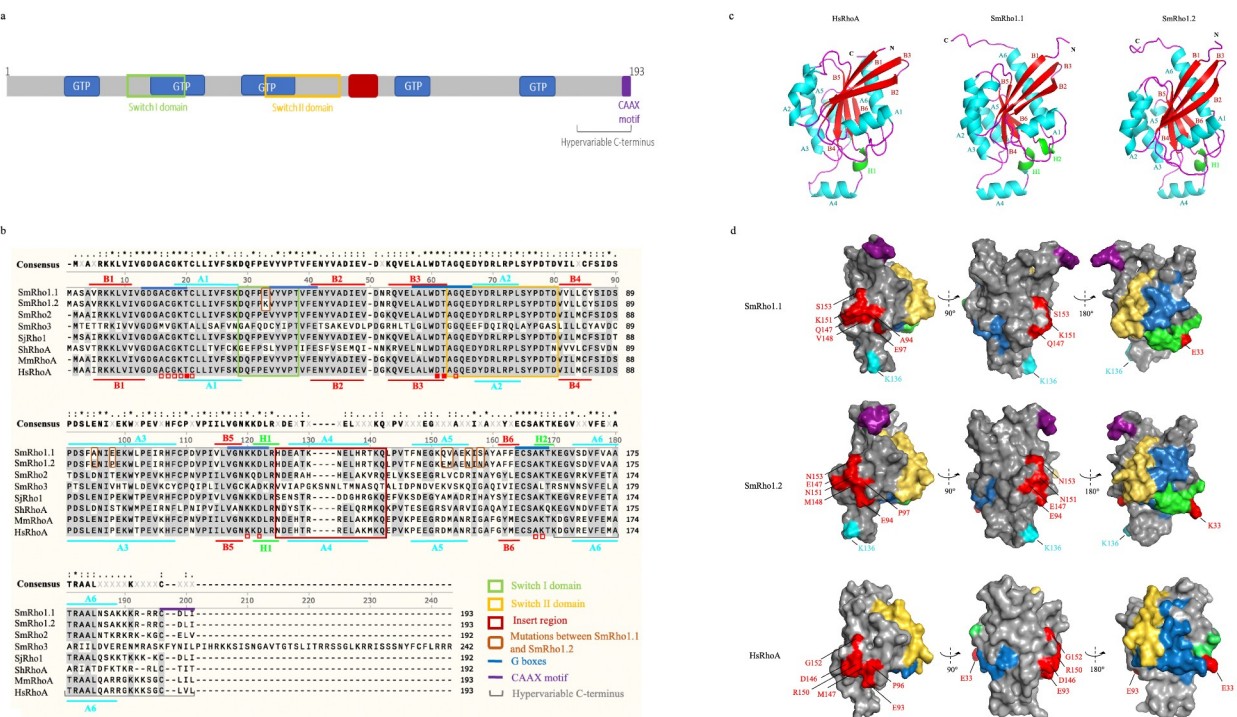

**Fig 1. Molecular and structural characterization of two SmRho1 isoforms.** (A) The SmRho1 isoforms are proteins of 193 aa. SmRho1.1 and SmRho1.2 contain five GTPase domains named "G boxes" (G1-G5) (in blue) and two functional elements, Switch I and II (boxed in green and yellow, respectively), which can interact with many regulators (GEFs, GAPs and GDIs) and effectors. The insert region (in red) is essential for Rho GTPase functions [37]. The C-terminal part presents a hypervariable domain and a prenylation motif CAAX (C = cysteine residue, A = aliphatic residue, X = terminal aa). (B) Sequence alignment between the *S. mansoni* Rho1 isoforms, *S. mansoni* Rho2 (Uniprot G4V9A8) and Rho3 (Uniprot Q8I8A0), *S. japonicum* Rho1 (Uniprot Q8MUI8), RhoA of *S. haematobium* (Uniprot A0A094ZFT0), human RhoA (Uniprot P61586) and mouse RhoA (Uniprot Q9QUI0). Sequences were aligned using ClustalW Multiple Alignment (SnapGene). The identical and semi-conserved aa are highlighted in black and gray respectively. Residues not conserved between SmRho1.1 and SmRho1.2 are boxed in orange. SmRho1.1 and SmRho1.2 both share high sequence similarity with HsRhoA (72% et 73% respectively) The secondary structure elements of SmRho1 isoforms and HsRhoA are indicated at the top and below the aligned sequences, respectively with A for α helices, B for β sheets and H for 3^10 helices. (C) Modeling of the tertiary structure of SmRho1 isoforms and HsRhoA [28]. Shown is a ribbon representation of SmRho1 isoforms and HsRhoA with β-strands (red), α-helices (cyan), and 310-helices (green). (D) Surface representation of proteins structures with Switch I and II (green and yellow, respectively), G boxes (in blue). For SmRho1 isoforms, the CAAX prenylated site is shown in purple and the acetylated lysine in cyan. The aa sequence differences between SmRho1 isoforms and HsRhoA are shown in red.

(G1-G5) (in blue) and two functional elements, Switch I and II (boxed in green and yellow, respectively), which can interact with many regulators (GEFs, GAPs and GDIs) and effectors (Fig 1A). SmRho1.1 and SmRho1.2 have an identical sequence in the switch I domain (residues 29–38 in RhoA) except for the positions 33 where the glutamic acid in SmRho1.2 is replaced by a lysine residue in SmRho1.1 (Fig 1A and 1B). As expected, the G domains and Switch II domain are completely conserved. At their C-termini both SmRho1.1 and SmRho1.2 contain a conserved prenylation (CDLI) CAAX motif (C represents cysteine, A is an aliphatic aa and X is a terminal aa) sequence. This motif is preceded by a run of basic aa (KKKRRR) which determine cellular localization for each protein [35], also typically found in Rho GTPases.

Using the threading method to predict SmRho1 isoform structures, we observed a conserved folding between SmRho1 isoforms and human RhoA consisting in a six-stranded βsheet surrounded by α-helices connected with loops, as found in human RhoA and other related small GTPases (Fig 1B and 1C). The β-sheet is formed by the anti-parallel association of two extended β-strands (B2 and B3) and the parallel association of five extended β-strands (B3, B1, B4, B5, B6) (Fig 1C). SmRho1.1 contains six α-helices (A1, A2, A3, A4, A5 and A6)

and two $3_{10}$-helices (H1–H2) whereas SmRho1.2 contains only one $3_{10}$ helix (H1) like human RhoA (Fig 1C). Three-dimensional models of the structures of SmRho1.1 and SmRho1.2 indicate that mutated aa (in red) are located on the surface of the proteins and form potential interaction domains for their partners (Fig 1D). The comparison with the HsRhoA three-dimensional model shows that differences in aa sequences do not affect the overall structure of this potential interaction surface (Fig 1D).

Several studies have resulted in the characterization of a number of Rho members subdivided into 8 subfamilies: Rho, Rac, Cdc42, RhoD/Rif, Rnd, Wrch-1/Chp, RhoH and RhoBTB [36] (Fig 2). In order to understand the phylogenetic relationships within the schistosome Rho family and within metazoans, we constructed a phylogenetic tree including the subfamilies of the Rho GTPases and Ras superfamily with mitochondrial Rho GTPases as an outgroup. Aa sequences from vertebrates, insects, nematodes, cestodes, trematodes and ascomycetes were included in this analysis. Phylogenetic analysis showed that schistosomes have a low number of orthologs of the main mammalian Rho subfamilies. We initially identified 7 Rho-like proteins in *S. mansoni*, but only 4 in *S. haematobium* and in *S. japonicum*. We found that schistosome Rho, Rac and Cdc42 clustered together with all Rho, Rac and Cdc42 orthologues (Fig 2, Red, green and purple clusters).

Maximum Likelihood (ML) analysis was also performed to characterize the phylogenetic positions of the recently discovered SmRho1.1 and SmRho1.2 Rho proteins. These proteins, cluster together with high fidelity (bootstrap = 99) inside the RhoA, B, C subfamily, indicating that *SmRho1.1* and *SmRho1.2* are probably paralogous genes that are orthologs to the human RhoA, B, C family. *S. haematobium* and *S. japonicum* do not seem to have undergone the same duplication, suggesting that the SmRho1 duplication is recent. Moreover, the phylogenetic analysis indicates probable conserved functions between the vertebrate and platyhelminth proteins (Fig 2, Red cluster).

## Biological functions of the two SmRho1 isoforms

Two independent Co-IP experiments were performed (named IP1 and IP2), using an anti-SmRho1 antibody which we produced in house in mice (S2 Fig) with pre-immune sera as control. MS analysis of the Co-IP proteins identified 1,000 different proteins (S3 Table). As expected, we demonstrated that our mouse anti-SmRho1 antibodies were not able to discriminate between the two SmRho1 isoforms (S1 Fig). We also noted a high degree of variation between identified proteins of each Co-IP experiment that can be explained by difficulties in trapping dynamic protein complexes, which potentially depend on post-translational modifications and GTP/GDP levels. In addition, *S. mansoni* is a complex multicellular parasite and protein quantities can vary between the different cellular types within a given worm as well as between different parasites. Finally, it should be noted that the two parasite protein extracts were each obtained from a pool of *S. mansoni* adult worms couples. Of the 1,000 proteins for which peptides were detected, we retained 86 and 32 proteins from IP1 and 2 respectively. Proteins that completed the three following criteria were considered: (i) at least three peptides in the Co-IP experiment, (ii) with no more than two peptides in the control and (iii) with a spectral count ratio between Co-IP SmRho1 and control of greater than 3.

Proteins identified as SmRho1 (SmRho1.1 Accession number Q9XZG7; Smp_330190; SmRho1.2 is not present among the current gene predictions in Wormbase Parasite and lacks an Smp identifier) interactors in either experiment are involved in 23 different biological processes (Fig 3A), the most abundant proteins corresponding to cytoskeleton organization (19%). Among the proteins identified, three are common between the two Co-IP experiments: cofilin (Accession number G4LZY0; Smp_195080.1), Putative Rho2 GTPase (Accession

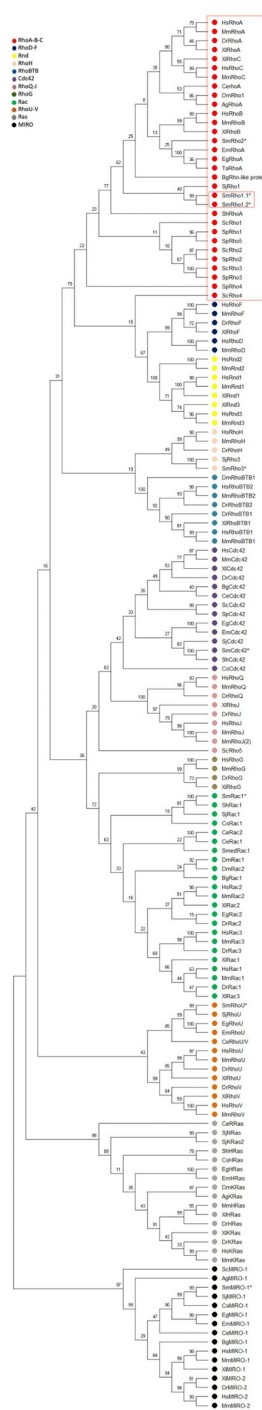

**Fig 2. SmRho1.1 and SmRho1.2 are orthologous to human RhoA.** Phylogenetic tree of Rho GTPases from vertebrates, platyhelminths (trematodes, cestodes and turbellarians), nematodes and insects, obtained with the ML algorithm. Numbers on internal branches are the bootstrap values. SmRho1 isoforms are circled in red. SmRho1.1 and SmRho1.2 cluster with the human RhoA, B, C clade (in red rectangle). The data generated also suggest that SmRho1.1 and SmRho1.2 are paralogs, which are orthologous to human RhoA, and originate from a recent gene duplication. The schistosome Rho GTPases are indicate with a *.

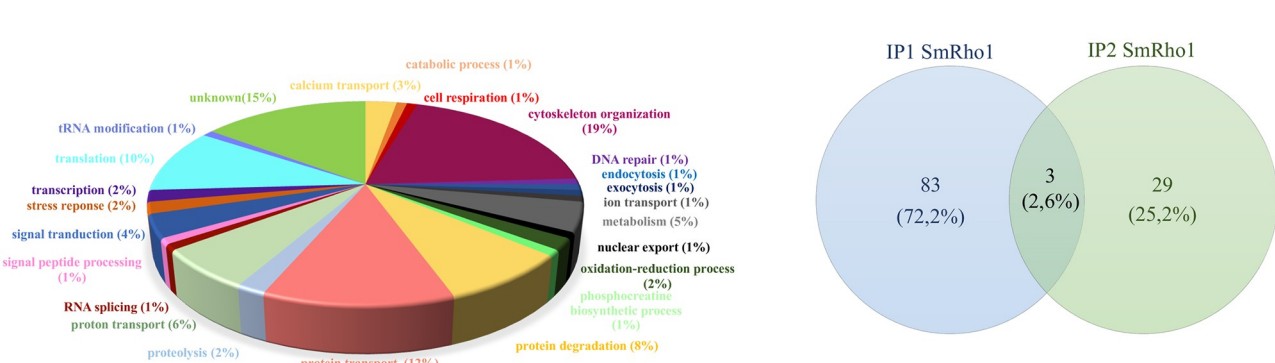

**Fig 3. Biological processes involving the proteins identified by Mass spectrometry as SmRho1 partners.** (A) Pie chart showing the percentage of involvement of the identified proteins and their biological processes. The processes were defined using the Blast2GO software. Two independent immunoprecipitation experiments were performed and grouped into one graphic display. (B) Venn diagram. Graphic shows common proteins between the two IP experiments (IP1 SmRho1 and IP2 SmRho1). The Venn diagram was established from Venny V2.1 [40].

number G4VBR2; Smp_072140) and Putative methylthioadenosine phosphorylase (Accession number G4VP86; Smp_171620) (Fig 3B).

Several regulatory partners were identified in co-IP with SmRho1, for instance the RhoGDI protein (Rho GDP-dissociation inhibitor-related, Accession number G4VK76; Smp_045610.1), a negative regulator of Rho GTPase activity [38], suggesting that this regulatory mechanism for Rho GTPase activity is conserved in *S. mansoni*. Other regulatory proteins like GEFs or GAPs were absent from the SmRho1 interactome in our experiments. However, we did detect an interaction between SmRho1 and SmRab6 (Putative Rab6, Accession number G4LXF1; Smp_163580) that could suggest the existence of a connection between the signaling of Rab proteins with SmRho1 during membrane trafficking, probably via the GEF regulator ARHGEF10, which in humans interacts with both Rab6A and Rab8A [39].

We have also identified Smkinesin (Kinesin-like protein, Accession number G4V5R8; Smp_001040), Smspectrin (Putative spectrin beta chain, Accession number G4VDE6; Smp_143470.1) and Smankyrin (Putative ankyrin, Accession number G4VKA7; Smp_145700) as SmRho1 partners. The cytoskeletal element, Smα-actinin (Putative alpha-actinin, Accession number G4VBW4; Smp_014780) was found in the SmRho1 interactome. α-actinins form a family of cytoskeletal actin-binding proteins playing crucial roles in organizing the framework of the cytoskeleton through crosslinking the actin filaments, as well as in focal adhesion maturation. In conclusion, our results show that SmRho1 is involved in cytosolic processes regulating cytoskeleton function, as is the human ortholog, HsRhoA. Despite the fact that we did not identify all the known protein partners of human RhoA, the identification of the different components of the cytoskeleton such as Smspectrin, Smα-actinin or Smcofilin suggests that signaling pathways involving SmRho1 are conserved in *S. mansoni*. However, since we cannot discriminate between the two SmRho1 isoforms using our MS analysis, no conclusions can be drawn concerning the specific roles of each SmRho1 isoform.

## SmRho1 interacts with SmHDAC8 and is acetylated on lysine K136 in *S. mansoni*

To confirm *in vivo* the direct interaction between SmHDAC8 and SmRho1 shown by Y2H screening (15), we performed CoIP experiments on adult worms (Fig 4A) and schistosomula

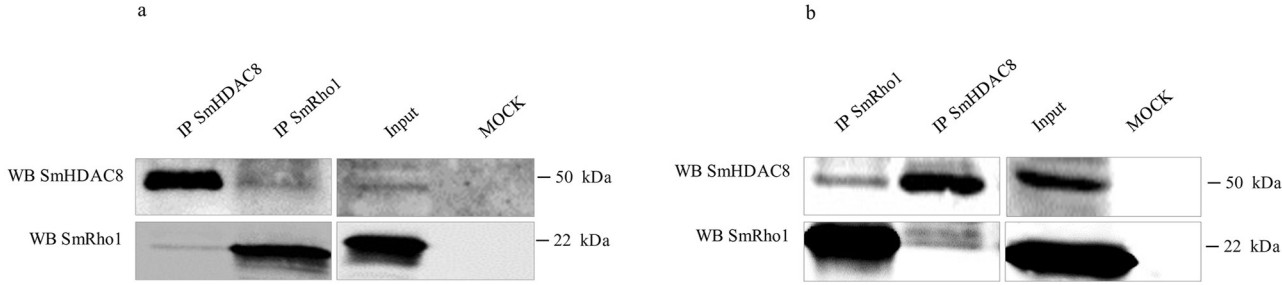

**Fig 4. SmHDAC8 interacts with SmRho1 in *S. mansoni* parasites.** Adult worms (A) and schistosomula (B) SmHDAC8 and SmRho1 were immunoprecipitated respectively using an anti-SmHDAC8 and anti-SmRho1 antibodies cross-linked to protein-L agarose beads. The immunoblots were probed with the same antibodies to detect the SmHDAC8 or SmRho1 protein in the input and eluates (labelled as IP SmHDAC8, IP SmRho1 or MOCK). As a negative control, we performed Co-IP (MOCK) with a mouse pre-immune serum alone in each experiment. The whole of IP eluate (60 μL) was loaded on SDS-PAGE. The results presented are from one experiment representative of three carried out.

(Fig 4B). Using anti-SmHDAC8 and SmRho1 antibodies, endogenous proteins were reciprocally and mutually immunoprecipitated and identified by Western Blotting. This result is consistent with protein-protein interaction between SmHDAC8 and SmRho1 in *S. mansoni* parasites (Fig 4). Nano LC-MS/MS analysis showing a mass increment of 42 Da corresponding to the presence of an acetyl group on lysine 136 of the TKQLPVTFNEGK peptide of SmRho1 was observed (S3 Fig). This may suggest that SmRho1 could be a substrate for SmHDAC8, but this remains to be investigated.

## SmRho1.1 C-terminal moiety binds SmHDAC8

To investigate the SmHDAC8–SmRho1isoforms interaction in more detail, we used the Y2H system in yeast and *X. laevis* oocytes as heterologous expression systems (Fig 5).

In Fig 5 (right panel), we show that only diploid yeasts which expressed SmHDAC8 and SmRho 1.1 could grow on the selective medium SD -Leu/-Trp/-His/-Ade, confirming an interaction between these two proteins. Moreover, the absence of growth for diploids expressing SmHDAC8 and SmRho1.2 suggested that the interaction between SmHDAC8 and SmRho1.1 was specific. We next co-expressed SmHDAC8 and SmRho1 isoforms in *X. laevis* oocytes (Fig 5B) to confirm the specific interaction between SmHDAC8 and SmRho1.1 (Fig 5A). We showed that Myc-tagged SmHDAC8 binds HA-tagged SmRho1.1 but not HA-tagged SmRho1.2.

Based on the specificity of interaction between SmHDAC8 and SmRho1.1 and in order to determine the segments of SmRho1.1 responsible for this interaction, various SmRho1 mutant proteins were produced by site-directed mutagenesis. Protein sequence alignments of SmRho1.1 and SmRho1.2 showed that there were only 7 different aa between the two isoforms (Fig 1B). These mutations are mainly located in the C-terminal part of the protein. In the N-terminal part, only the glutamic acid at position 33 of SmRho1.1 is substituted by a lysine in SmRho1.2. Two N-terminal fragments of different sizes were therefore produced, by insertion of premature stop codon, using SmRho1.1 in order to identify the domain binding to SmHDAC8 (Fig 5C). In parallel, two other mutant proteins were produced using both SmRho1.1 and SmRho1.2 as templates. The glutamic acid at position 33 of SmRho1.1 was substituted by a lysine (SmRho1.1 E33K). Similarly, the lysine at position 33 of SmRho1.2 was replaced by a glutamic acid (SmRho1.2 K33E) (Fig 5C). We co-expressed SmHDAC8 and SmRho1 mutants in *X. laevis* oocytes (Fig 5D) and showed that Myc-tagged SmHDAC8

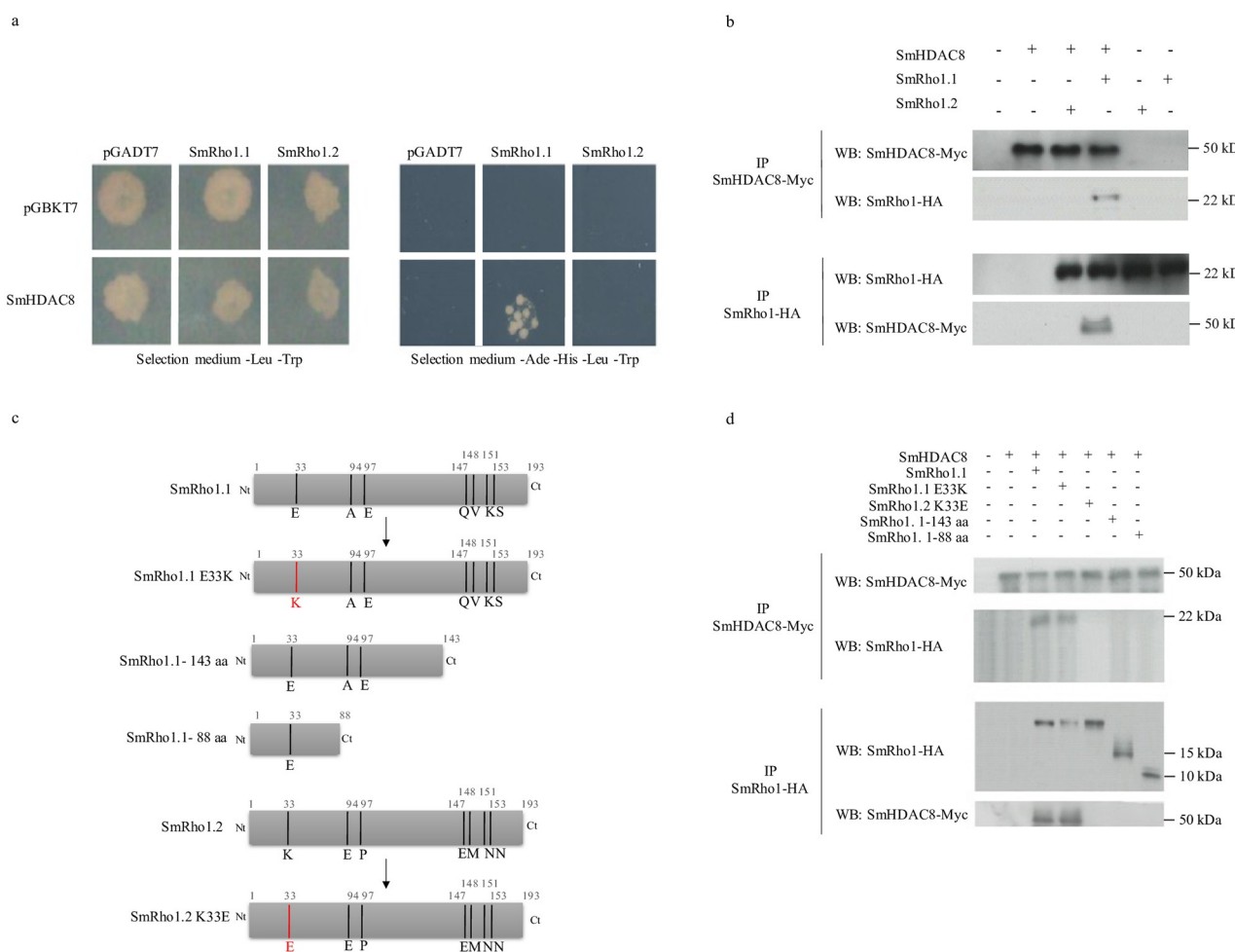

**Fig 5. The interaction between SmRho1.1 and SmHDAC8 is dependent on the SmRho1.1 C-terminus.** (A) Y2H mating experiments showed that SmHDAC8 interacts specifically with SmRho1.1 protein. AH109 yeasts expressing only Gal4AD (pGADT7) or Gal4AD-fused SmRho1.1 or SmRho1.2 were mated with Y187 yeasts expressing only Gal4DBD (pGBKT7) or Gal4DBDfused SmHDAC8. Diploids were allowed to grow on a minimal SD -Leu/-Trp medium (left panel) and diploids expressing interacting proteins were then selected on SD -Leu/-Trp/-His/Ade medium (right panel). Only yeasts expressing SmHDAC8 and SmRho1.1 grew on the selective medium. The results presented are from one experiment representative of three carried out. (B) SmHDAC8 binds only SmRho1.1 but not SmRho1.2. Co-IP and WB analysis of SmHDAC8 and SmRho1 isoforms expressed in *X. laevis* oocytes showed an interaction only between SmHDAC8 (Myc-tagged) and SmRho1.1 (HA-tagged). cRNAs encoding HA-tagged SmRho1.1 or SmRho1.2 were co-injected in *X. laevis* oocytes with cRNA encoding Myc-tagged SmHDAC8. Oocytes were incubated in ND96 medium and lysed. Proteins from soluble extracts were immunoprecipitated by anti-HA or anti-Myc antibodies and analyzed by WB to detect SmHDAC8- Myc (50 kDa) and SmRho1-HA isoforms (22 kDa) with anti-Myc or anti-HA antibodies. Experiments were repeated three times on oocytes from three different females. (C) Schematic structure of SmRho1.1 and SmRho1.2 mutants. Using site-directed mutagenesis, the glutamic acid Glu33 of SmRho1.1 was substituted by a lysine (SmRho1.1 E33K) and the lysine Lys33 of SmRho1.2 by a glutamic acid (SmRho1.2 K33E). SmRho1. 1–143 aa and SmRho1. 1–88 aa proteins are portions of SmRho1.1. produced by site-directed mutagenesis. (D) Co-IP and WB experiments performed in X. laevis oocytes revealed that SmRho1. 1–143 aa and SmRho1. 1–88 aa mutants (HA-tagged) are not able to bind SmHDAC8 (Myc -tagged). cRNAs encoding HA-tagged SmRho1 isoforms, SmRho1.1 mutant or SmRho1.2 mutants were co-injected in X. laevis oocytes with cRNA encoding Myc-tagged SmHDAC8. Oocytes were incubated in ND96 medium and lysed. Proteins from soluble extracts were immunoprecipitated (IP) by anti-HA or anti-Myc antibodies and analyzed by WB to detect SmHDAC8 (50 kDa), SmRho1 isoforms (22 kDa) or SmRho1 mutants (22kDa) with anti-Myc or anti-HA antibodies. SmHDAC8 co-immunoprecipitated only with SmRho1.1 E33K.

interacts with HA-tagged SmRho1.1E33K but not with SmRho1.2 K33E. These substitutions therefore failed to change the binding behavior, compared to the SmRho1 wild type isoforms.

In addition, we showed that HA-tagged SmRho1 N-terminal fragments cannot interact with Myc-tagged SmHDAC8 (Fig 5D) suggesting that the SmRho1.1 C-terminal moiety is

involved in the binding to SmHDAC8. However, it is also possible that the SmRho1.1 fragments could be misfolded, inducing a loss of protein function and their ability to interact with SmHDAC8. In consequence, we performed point mutations using SmRho1.1 and SmRho1.2 to identify specific residues involved in the interaction with SmHDAC8. Using site-directed mutagenesis, the glutamine Q147 and the valine V148 of SmRho1.1 were substituted by a glutamic acid and a methionine (SmRho1.1 EM) and then the lysine K151 and the serine S153 by two asparagines (SmRho1.1 EMNN) (S6A Fig). We also produced SmRho1.2 QV and SmRho1.2 QVKS mutants by site-directed mutagenesis using the SmRho1.2 protein as template (S6A Fig). We then co-expressed the resulting mutant proteins in *X. laevis* oocytes. CoIP experiments revealed that none of the SmRho1.1 and SmRho1.2 mutants (HA-tagged) were able to bind SmHDAC8 (Myc-tagged) (S6B Fig). In conclusion, the mutations we carried out to SmRho1.2 were incapable of restoring the interaction between SmHDAC8 and SmRho1.2, suggesting that all seven aa differentiating the two isoforms are potentially involved in the interaction with SmHDAC8.

## SmRho1.2 interacts specifically with SmDia

Because we have shown a specific interaction between SmHDAC8 and SmRho1.1 in yeast and *X. laevis* oocytes suggesting that each isoform could have specific partners, we investigated the interaction between SmRho1 isoforms and the Rho Binding Domain of SmDia (SmDia-RBD) (24) using the Y2H system in yeast and *X. laevis* oocytes.

In Fig 6A, we show that only diploid yeasts that express SmRho1.2 and SmDia-RBD grow on the selective medium SD -Leu/-Trp/-His/-Ade, indicating an interaction between these two proteins. In order to confirm the specific interaction between SmRho1.2 and SmDia-RDB (Fig 6B), we co-expressed SmRho1 isoforms and SmDia-RBD in *Xenopus* oocytes. This shows that Myc-tagged SmDia-RBD specifically binds HA-tagged SmRho1.2.

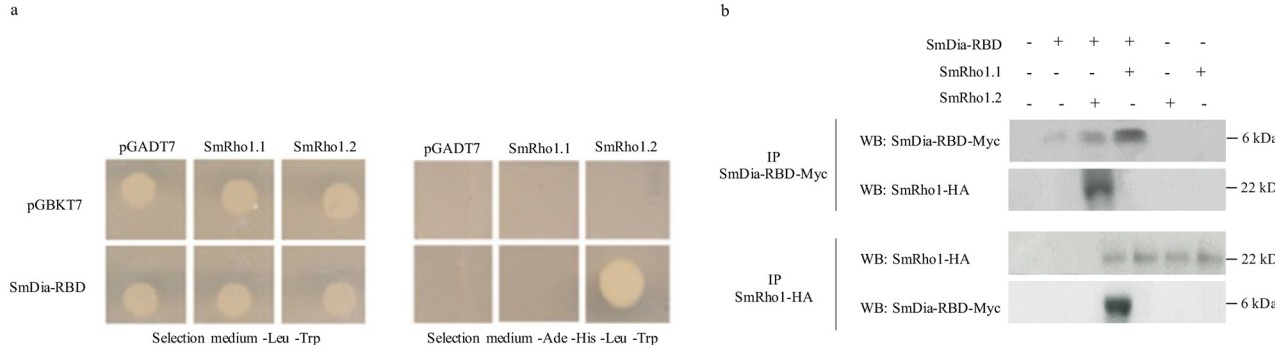

**Fig 6. SmDia binds SmRho1.2 but not SmRho1.1.** (A) Y2H mating experiments showed that SmDia-RBD (R̲ho B̲inding D̲omain) interacts specifically with SmRho1.2 protein. AH109 yeasts expressing only Gal4AD (pGADT7) or Gal4AD-fused SmRho1.1 or SmRho1.2 were mated with Y187 yeasts expressing only Gal4DBD (pGBKT7) or Gal4DBD-fused SmDia-RBD. Diploids were allowed to grow on a minimal SD -Leu/-Trp medium and diploids expressing interacting proteins were then selected on medium. Only yeasts expressing SmDia-RBD and SmRho1.2 grew on the SD -Leu/-Trp/-His/-Ade selective medium. The results presented are from one experiment representative of three carried out. (B) SmDia-RBD interacts with SmRho1.2. Co-immunoprecipitation and WB analysis of SmDia-RBD and SmRho1 isoforms expressed in X. laevis oocytes showed an interaction only between SmDia-RBD (Myc-tagged) and SmRho1.2 (HA-tagged). cRNAs encoding HA-tagged SmRho1.1 or SmRho1.2 were co-injected in oocytes with cRNA encoding Myc-tagged SmDia-RBD. Oocytes were incubated in ND96 medium and lysed. Proteins from soluble extracts were immunoprecipitated (IP) by anti-HA or anti-Myc antibodies and analyzed by WB to detect SmDia-RBD (6 kDa) and SmRho1 isoforms (22 kDa) with anti-Myc or anti-HA antibodies. Experiments were performed three times on oocytes from three different females.

## SmHDAC8 inhibition or knockdown cause disruption of the parasite actin cytoskeleton

Finally, to elucidate the role of SmHDAC8 in the regulation of cytoskeleton dynamics, we examined the impact of SmHDAC8 inhibition on the organization of the actin network of the parasite tegument using both RNAi and selective inhibitors (Figs 7 and 8).

Adult worms and schistosomula were first treated with a selective SmHDAC8 inhibitor, TH65 (33), then fixed and Actin F was stained with Alexa488 conjugated phalloidin (green) and nuclei with DAPI (blue) (Figs 7 and 8, S4 and S5 Figs). As a reference, we used parasites treated with Trichostatin A (TSA) which inhibits both class I and II mammalian histone deacetylases (Figs 7A and 8A and S4 Fig). In parallel, we used an RNAi complementary approach to target transcripts encoding SmHDAC8 (Figs 7B and 8C and S5 Fig).

In adult worms, the TH65 inhibitor did not induce a significant disorganization of tegumental actin (Fig 7A and S4 Fig). Indeed, phalloidin, which binds to actin filaments, was detected in spines and tegumental cells and also in subtegumental muscle fibers. Actin filaments appeared as horizontal and vertical straight lines stretching across the whole thickness of the tegumental syncytium (Fig 7A and S4 Fig). We could, however, observe that TH65 seems to impact the structure of the spines. (Fig 7A and S4 Fig). Moreover, no significant effects were observed in adult worms after treatment with TSA (Fig 7A and S4 Fig) or *Smhdac8* interference (Fig 7B and S5 Fig).

On the contrary, schistosomula treated with TH65 and TSA were highly affected at various levels and, for some of the parasites, we observed a strong phenotype with defective muscular actin organization (Fig 8A). In control parasites (Fig 8A: DMSO section, top and bottom panel), phalloidin staining revealed higher-order actin structures, forming a network which seems to correspond to the muscle fibers of the subtegumental muscle layer. In treated parasites (Fig 9A and 9B, TH65 and TSA section, top and bottom panel), we observed that the actin network structure was disrupted after inhibitor treatment with a modification of actin filament structure or absence of F-actin.

While dsRNA promoted only 22% silencing of the *SmHDAC8* gene, in schistosomula this nevertheless led to a significant effect on the actin cytoskeleton organization (Fig 8B and 8C), similar to that observed after inhibitor treatment (Fig 8B). This suggests that the effects of TH65 could be due to selective inhibition of SmHDAC8.

In order to confirm the involvement of SmHDAC8 in the SmRho1 signaling pathway, we treated parasites with Rho inhibitor I (Figs 7A and 8A), used to selectively inactivate the human GTPases RhoA, RhoB, and RhoC by ADP-ribosylation on asparagine 41, which is conserved in SmRho1 isoforms, and we used RNAi to knock down *SmRho1* (Figs 7B, 7C and 8B). In adult worms, inhibition and silencing of *SmRho1* did not significantly affect actin network organization, but in schistosomula we observed a very similar phenotype to that obtained after inhibition and silencing of *SmHDAC8*.

Taken together, these findings suggest that SmHDAC8 is involved in regulation of actin cytoskeleton organization in *S. mansoni*, more evidently in schistosomula. However, the robust silencing and inhibition of SmHDAC8 in adult worms did not result in any significant phenotypic changes. This may be due to a more active turnover of the actin cytoskeleton in the larvae compared to adult worms. It should be noted that TH65 causes inhibition of the deacetylation activity of SmHDAC8 enzyme, but does not affect protein expression.

## Discussion

In the present study, we have confirmed SmRho1 as a partner and potential substrate of SmHDAC8 and provided evidence that the latter is involved in the regulation of the actin cytoskeleton in *S. mansoni*.

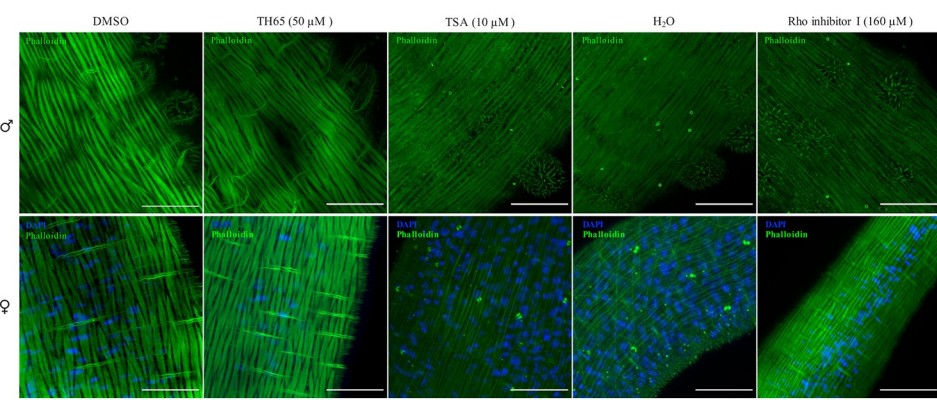

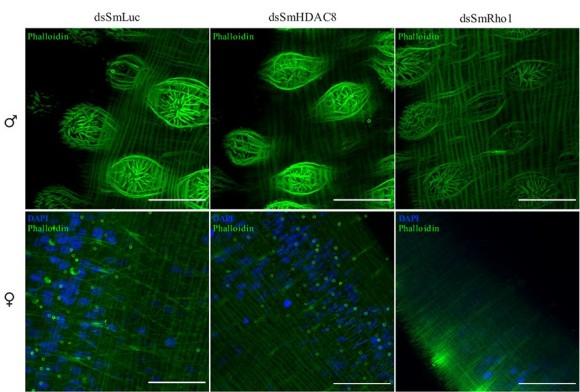

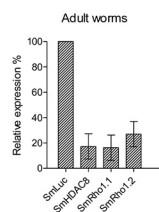

**Fig 7. Effect of SmHDAC8 inhibition on actin filament of *S. mansoni* adult worms.** (A) Effect of SmHDAC8 inhibition in male and female adult worms. Freshly perfused adult couples were maintained in culture for 16 hours and incubated with DMSO or with 50 μM of TH65 or 10 μM of TSA, then fixed and stained with phalloidin (green) and DAPI (blue). $H_2O$ and DMSO were used as negative controls. As a positive control, schistosomula were treated with a Rho inhibitor I (160 μM). Scale bar represents 20 μm, magnification, x630. Experiments were performed three times on adult worms obtained by three different hepatic portal perfusions of hamsters. (B) Effect of *SmHDAC8* transcript knockdown in adult worms. RNA interference was carried out by *S. mansoni* worms with dsRNA for *SmHDAC8*, *SmRho1* (positive control) *or luciferase* (negative control) as described in the Methods section. Actin-F was revealed with phalloidin staining and the nuclei were stained with DAPI (blue). Microscopic examination was carried out 5 days after RNAi treatment. Scale bar represents 20 μm, magnification, x630. Experiments were performed three times on adult worms obtained by three different hepatic portal perfusions of hamsters. (C) RT-qPCR results of RNAi treatment with dsRNA for SmHDAC8 (dsSmHDAC8), SmRho1 (dsSmRho1) or luciferase (dsSmLuc) and analyses of relative transcript levels of *SmHDAC8*, *SmRho1.1* or *SmRho1.2* in adult worms. *SmPSMB7* was used as an internal reference gene (32). The results were analyzed using the $2^{-\Delta\Delta CT}$ method and experiments were carried out once in duplicate.

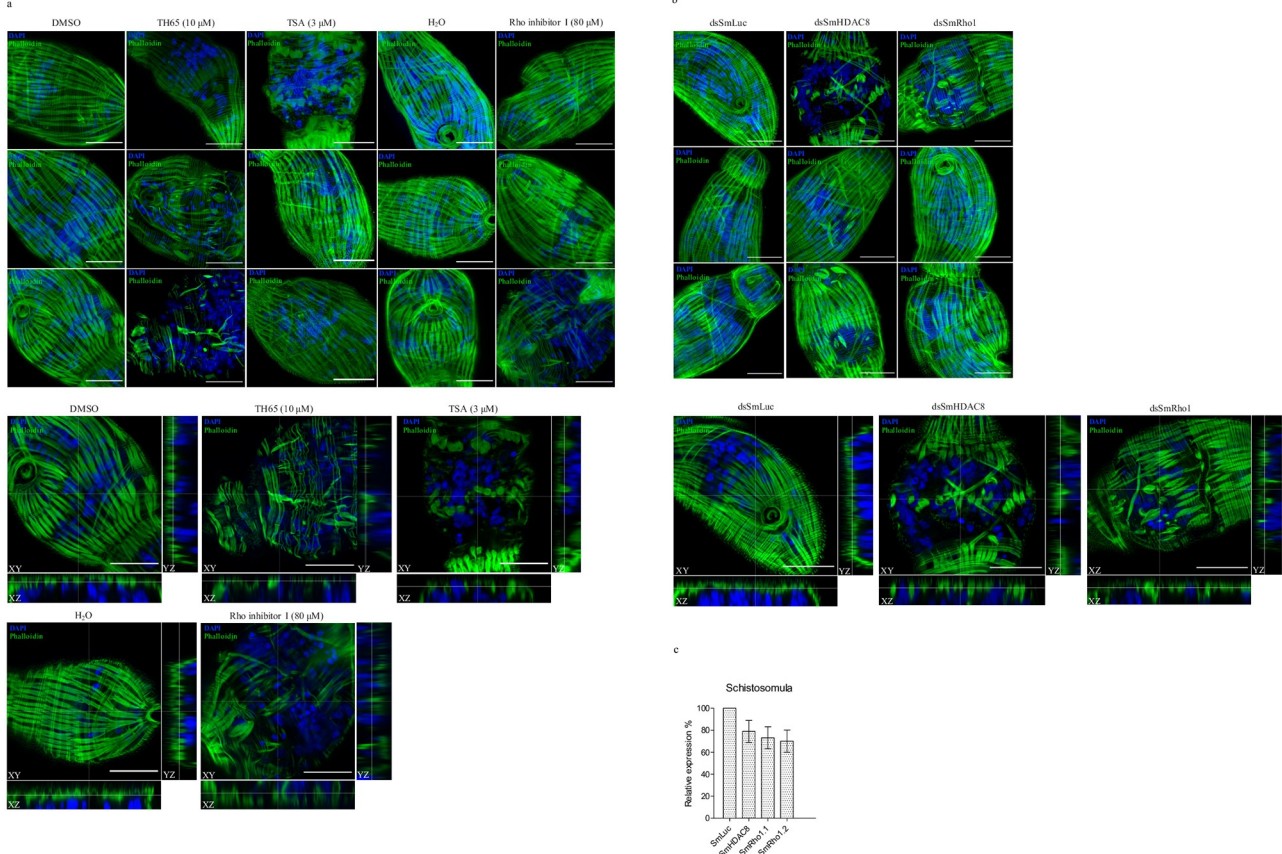

**Fig 8. Effect of SmHDAC8 inhibition on actin filaments of *S. mansoni* schistosomula.** (A) Effect of SmHDAC8 inhibition in schistosomula. Airyscan microscopy images taken of schistosomula treated for 16 hours with a SmHDAC8 selective inhibitor (TH65), at 50 μM and a pan-inhibitor (TSA) at 3 μM. Parasites treated with DMSO or H$_2$O were used as negative controls. As a positive control, schistosomula were treated with a Rho inhibitor I (80 μM). Actin-F was revealed with phalloidin staining. The nuclei were stained with DAPI (blue). Results shown are representative of three independent experiments. Scale bars represent 20 μm, magnification, x630 (top panel). Airyscan images with orthogonal views of treated *S. mansoni* schistosomula are also shown. Results shown are from one experiment. Scale bar represents 20 μm, magnification, x630 (bottom panel). (B) Effect of SmHDAC8 transcript knockdown in schistosomula. RNA interference was carried out by schistosomula with dsRNA for SmHDAC8, SmRho1 (positive control) or luciferase (negative control) as described in the Methods section. Actin-F was revealed with phalloidin staining and the nuclei were stained with DAPI (Blue). Microscopic examination was carried out 2 days after RNAi treatment. Scale bar represents 20 μm, magnification, x630 (top panel). Airyscan images with orthogonal views of *S. mansoni* schistosomula are also shown. Results shown are from one experiment. Scale bars represent 20μm, magnification, x630 (bottom panel). (C) RT-qPCR results of RNAi treatment with dsRNA for SmHDAC8 (dsSmHDAC8), SmRho1 dsSmRho1) or luciferase (dsSmLuc) and analyses of relative transcript levels of SmHDAC8, SmRho1.1 or SmRho1.2 in schistosomula. SmPSMB7 was used as an internal reference gene (32). The results were analyzed using the 2−ΔΔCT method and experiments were carried out once in duplicate.

Overall, the MS approach led to the detection of a significant number of proteins ortholo-gous to human proteins that are involved in the regulation of the organization of the cytoskele-ton. The identification of SmRho2 as a partner of SmRho1 is unexpected (S3 Table). It is possible that the presence of SmRho2 in the SmRho1 interactome can be explained by the IP of this isoform using anti-SmRho1 antibodies, probably due to the significant sequence iden-tity between SmRho1 and SmRho2 (72%) (19).

We did not identify SmROCK, GEFs and GAPs, and more surprisingly, SmHDAC8 as SmRho1-interacting proteins. As we have already seen, the dynamics of protein–protein inter-action networks or absence of post-translational modifications could explain the absence of the histone deacetylase. Moreover, some proteins identified may be members of immunopre-cipitated complexes and are not direct partners and it is possible that our polyclonal serum

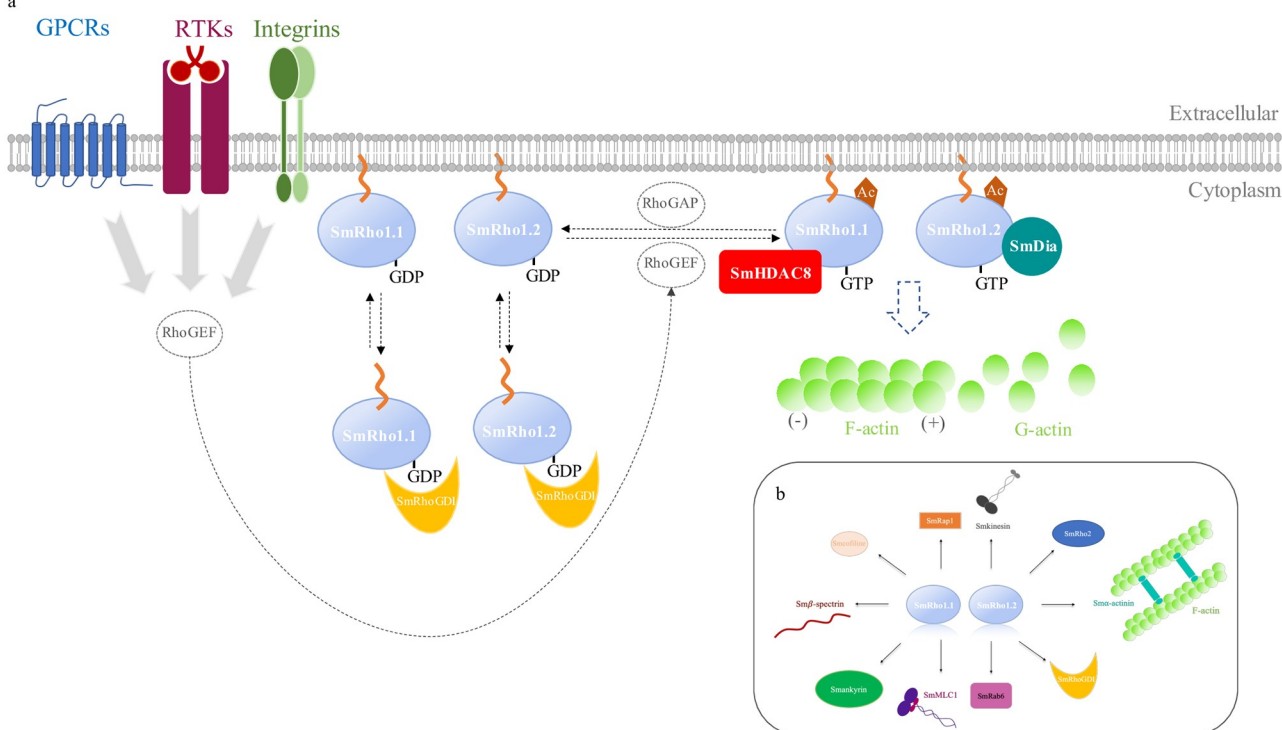

**Fig 9. Model of signaling pathways involving SmHDAC8 and SmRho1 isoforms in cytoskeleton organization based on the human RhoA Signaling Pathway.** A model summarizing the different pathways putatively regulated by the SmRho1 isoforms is shown in Fig 9. In this model, SmRho1.1 and SmRho1.2 would impact on different pathways of cytoskeleton organization via interaction respectively with SmHDAC8 and SmDia. A) Schematic model showing proteins involving in cytoskeletal events in *S. mansoni* adult worms and schistosomula. We propose the existence of two signaling pathways in the parasite involving the two SmRho1 isoforms. Various extracellular stimuli trigger the activation of RhoGEFs that catalyze the exchange of GDP for GTP to activate SmRho1.1 and SmRho1.2. Activated SmRho1.1 and SmRho1.2 bind to SmHDAC8 and SmDia respectively that transmit Rho mediated signals to downstream target proteins which will regulate actin polymerization. Proteins encircled in grey (RhoGEF and RhoGAP) are absent from the SmRho1 interactome. (B) An overview of SmRho1-interacting proteins.

masked the binding sites with some of them. In support of this hypothesis, Karolczak-Bayatti and coworkers have shown that human HDAC8 co-immunoprecipitates with cofilin [41]. Human cofilin is involved in the stabilization of actin filaments and forms a protein complex with RhoA and ROCK [37]. Although ROCK was not found as a partner of SmRho1, it is comforting to find cofilin as one of the members of a potential parasite protein complex formed by SmRho1, SmROCK1, SmLIMK and the Myosin light chain (MLC, Accession number G4VBS3; Smp_045220) also identified as an SmRho1 partner. We can assume that SmHDAC8 and SmRho1 form a multiprotein complex, especially with cofilin, which we have identified as SmRho1 partner. Moreover, in humans ROCK1 is the effector of GTP-RhoA and after its activation, the downstream effectors, myosin light chain (MLC) and cofilin are activated by phosphorylation. Phosphorylated MLC stimulates the binding of myosin to actin to regulate stress fiber formation and actomyosin contractility [42]. It was also shown that the activity of RhoA and its effector ROCK is inhibited by Rap1 signaling and PKA to regulate coronary artery relaxation [43]. The connection between SmRho1 and SmRap1 (Putative Rap1, Accession number G4VE67) illustrates once again the involvement of the GTPase in the regulation of cytoskeletal events.

The presence of actin binding proteins and regulators of the actin cytoskeleton like spectrin or ankyrin in the SmRho1 interactome, provides additional clues concerning a signaling pathway dependent on SmRho1. Intriguingly, some, such as α-actinin, and spectrin interact

directly or indirectly with NMDA receptors [44]. Moreover, p250GAP is the first Rho family GAP GTPase shown to be enriched in the NMDA receptor complex, suggesting that p250GAP links the NMDA receptor to actin reorganization via RhoA [45].

Kinesin was also found in the SmRho1 interactome. Kinesins belongs to a class of motor proteins that move along microtubules. Pan and coworkers found that BNIP-2, a BCH domain containing protein binds RhoA, RhoGEF and kinesin-1 to regulate microtubule dynamics [46]. In addition, it was shown that RhoA-GTP binds a protein inserted in endoplasmic reticulum membranes named kinectin or KTN1 that interacts with the cargo-binding site of kinesin, thus activating its microtubule-stimulated ATPase activity, which is required for vesicle motility [47]. Kinesin also interacts with spectrin and ankyrin to regulate intracellular organelle transport [48]. In eukaryotic cells, spectrin is a cytoskeletal protein that lines the intracellular side of the plasma membrane. It forms pentagonal or hexagonal arrangements, forming a scaffold that plays an essential role in plasma membrane integrity and cytoskeletal structure. Ankyrins form a family of proteins that mediate the attachment of membrane proteins to the spectrin-actin cytoskeleton. Thus, the ankyrin–spectrin assembly provides mechanical stability to the lipid bilayer in addition to organization of membrane proteins. Moreover, in *X. laevis*, Cho and coworkers demonstrated that cytoskeletal organization, cell adhesion and ectodermal integrity are biological processes regulated by a catenin-spectrin-ankyrin-p190RhoGAP complex via RhoA activity [49].

To support our data of SmRho1interactome, we used single-cell RNAseq data of Collin's lab as additional support (http://www.collinslab.org/schistocyte/) and we have noted that some of SmRho1 partners share the same compartments/organs for expression. We found that *SmRho1* (Smp_330190) is co-expressed with *SmROCK1* (Smp_139800) in parenchyma and muscles. In *S. mansoni* tegument, Smkinesin (Smp_001040) and Smα-actinin (Smp_014780) are expressed with *SmRho1*. Moreover, expression levels of these genes appear to be influenced by pairing and they are highly expressed in tegument, parenchyma and muscles of females, and specially particularly in mature females. Interestingly, in each compartment in males or females, we identified genes encoding for specific RhoGEFs and RhoGAPs which are co-expressed with *SmRho1*. For instance, in parenchyma of matures females, we found genes encoding for two RhoGAPS (Smp_138150, Smp_122680) and one for a RhoGEF (Smp_316340).

Another element to be kept in mind is that SmRho1 may be a substrate of SmHDAC8, leading to an ephemeral interaction. Moreover, SmHDAC8 is not highly expressed in adult worms [11]. CoIP followed by Western Blotting, which is more sensitive, did allow us to detect SmHDAC8 bound to SmRho1. In addition, this interaction was confirmed *in vitro* using the proteins overexpressed in yeast and *X. laevis* oocytes.

The use of selective inhibitors of SmHDAC8 and SmRho, as well as knockdown of their transcripts using RNAi, strongly suggests that both proteins are involved in maintaining the integrity of the actin cytoskeleton. Because SmHDAC8 has multiple interactants, we can nevertheless not conclude that the observed phenotypes are the consequence of deregulation of a signaling pathway jointly mediated by SmRho1 and SmHDAC8. Indeed, an identified HsHDAC8 substrate is cortactin which contributes to the organization of the F-actin cytoskeleton. It was shown that cortactin-actin interaction is regulated by (de)acetylation and HsHDAC8 seems to influence smooth muscle contraction [21, 50]. Further to this point, we cannot rule out an effect of inhibition/knockdown of SmHDAC8 directly on the muscle cells themselves, leading to subsequent disruption of the cytoskeleton. This point merits further investigation.

In our experiments inhibition or transcript knockdown of SmHDAC8 or SmRho1 the effects observed were much more apparent in schistosomula than in adult worms. This result

apparently conflicts with our previous results showing the toxicity of both the pan-HDAC inhibitor TSA [12] and the selective SmHDAC8 inhibitor TH65 [14] for adult worms as well as for schistosomula. In the present study we developed a strategy designed to detect the early effects of inhibition or transcript knockdown on the parasite, whilst minimizing overt necrosis/apoptosis. The low level of effects on the adult worms, limited to the structure of the spines may be due to a relatively reduced access of the inhibitors to the worms compared to the larvae. However, the high level of transcript knockdown obtained for adult worms did not give rise to overt effects on the actin cytoskeleton, whereas a lower level of knockdown did cause extensive disruption of the cytoskeleton in schistosomula. This may suggest that schistosomula are particularly susceptible to interference into the maintenance of the cytoskeleton.

In future work we will consider determining the impact of inhibition of SmHDAC8 on the microtubular network and perform staining experiments after SmHDAC8 inhibition and KO using TH65 and RNAi. In 2011, Yamauchi et al. showed that the interference of transcripts encoding human HDAC8 using siRNA disrupted the microtubule network of cells [51].

Following our initial identification of SmRho1 as a potential partner for SmHDAC8, we have consolidated the demonstration of this interaction, both *in vitro* and within the parasite using co-immunoprecipitation studies. Our data demonstrated that SmHDAC8-SmRho1 interaction involves the C-terminal domain of SmRho1.1. However, our attempts to "transform" SmRho1.2 into SmRho1.1 *via* limited site-directed mutagenesis were unsuccessful, and suggest that the entire interaction interface formed by the aa that differ between the two isoforms is responsible for this difference in interaction.

To confirm this specificity of interaction it will be necessary in a future study to determine the interaction between SmHDAC8 and SmRho1 isoforms after IP with SmHDAC8 using adult worms and schistosomula protein extracts. The SmRho1 protein thus isolated will be analyzed in MS to identify which SmRho1 isoform is present. The observation that only SmRho1.1 can bind SmHDAC8, and that only SmRho1.2 can bind SmDia, suggests that these isoforms have developed distinct functions. This implies that there are two very distinct pathways that participate in the organization of the cytoskeleton *via* the two isoforms of SmRho1. Thus, only the SmRho1.1 isoform should interact with SmROCK for example, but this requires further investigation.

Moreover, although our results argue for a direct interaction between the SmRho1 isoforms and its partners, it should be borne in mind that the effects of inhibition and RNAi observed in the parasite may reflect the involvement of multiprotein complexes. Nevertheless, the detection of the acetylation of SmRho1 raises the possibility that it is a substrate of SmHDAC8 and that acetylation, could be involved in the modulation of the properties of SmRho1.

Although human RhoA and other Rho GTPases are not acetylated, Hong and coworkers identified an acetylation of *S. japonicum* Rho1 on lysine 141 [23]. This residue K141 is absent in SmRho1, and another target residue K136 in SmRho1 is apparently not acetylated in SjRho1. It should also be borne in mind that only one of the two isoforms of SmRho1 may be acetylated. Interestingly, there are regulatory proteins that could be described as "atypical regulators" and SmHDAC8 could be one of these. The Memo protein, for example, interacts with RhoA and appears to promote its membrane localization and therefore its activation, within a Memo-RhoA-mDia1 multiprotein complex [52]. Hence SmHDAC8 could also be considered as an atypical regulator of SmRho1 either via direct binding or *via* deacetylation. In addition, if only one of the two isoforms of SmRho1 is acetylated, the specific acetylation of SmRho1.1 and the potential deacetylation by SmHDAC8 could constitute a regulatory mechanism of the parasite-specific SmRho1-mediated signaling pathway. However, the lysine that we have identified as an acetylation site on SmRho1 is conserved in both isoforms suggesting that SmRho1 isoforms could be acetylated and deacetylated by the same KDAC. Tools like CRISPR-Cas9, would allow

obtaining conditional KO or KO parasites in order to study the specific role of *SmRho1* genes at different parasitic stages. Although this is not currently possible, projects concerning the development of CRISPR-Cas9 technology in schistosome are underway [53]. Finally, although SmDia interacts specifically with SmRho1.2, we have no knowledge of how the SmRho1.2-SmDia complex could participate in the regulation of the polymerization of actin filaments. However, in 2009, Quack and coworkers [24] demonstrated that SmDia is able to interact directly with the SmTK3 protein (Src -like Tyrosine-Kinase). Moreover, in humans, the RhoA-GTP/mDia/Scr complex is known to regulate the formation of actin filaments [54, 55].

## Supporting information

**S1 Fig. Identification of SmRho1.1 and SmRho1.2 by mass spectrometry following the IP of SmRho1 isoforms in adult worms.** The table indicates the sequence, the molecular weight, the percentage of sequence coverage and the number of "unique peptides/spectra" for each identified protein in two biological independent assays (IP1 and IP2).
(TIF)

**S2 Fig. Mouse antiserum anti-SmRho1 evaluation.** (A) Detection of adult worms endogenous SmRho1 and SmRho1.1 recombinant protein. The blots were probed with mouse prebleed sera, with mouse anti-SmRho1 antisera tested at 1/200, 1/500 and 1/1 000 dilution and with mAb anti-His. (B) Immunoblot analysis of recombinant SmRho1.1 (SmRho1.1-rec) and SmRho1.2 (SmRho1.2-rec) proteins with mouse anti-SmRho1 antisera. (C) detection of endogenous SmRho1 in total proteins extracted from parasitic stages of *S. mansoni*. (D) Immunodepletion of SmRho1antibodies using SmRho1.1 recombinant protein show the specificity of the antibody against SmRho1. Adult worms and schistosomula endogenous SmRho1 proteins were detected by WB using mouse anti-SmRho1 antisera (left panel) or mouse anti-SmRho1 depleted antisera (middle panel). Human RhoA antibody was used as negative control (right panel).
(TIF)

**S3 Fig. SmRho1 is acetylated on K136.** MS/MS spectrum of the acetylated peptide TK(Ac) QLPVTFNEGK. B (red) and y (blue) ion fragments series clearly identified peptide sequence and acetylation on lysine 136 with a Mascot ion score of 60.6. The experiment was performed twice on adult worms obtained by three different hepatic portal perfusions of hamsters.
(TIF)

**S4 Fig. Airyscan images with orthogonal views of *S. mansoni* adult worms after inhibitor treatment.** Scale bar represents 20 μm, magnification, x630. Experiments were performed three times on adult worms obtained by three different hepatic portal perfusions of hamsters.
(TIF)

**S5 Fig. Airyscan images with orthogonal views of *S. mansoni* adult worms after RNAi.** Scale bar represents 20 μm, magnification, x630. Experiments were performed three times on adult worms obtained by three different hepatic portal perfusions of hamsters.
(TIF)

**S6 Fig. SmRho1.2 mutations do not restore the interaction between SmHDAC8 and SmRho1.2.** (A) Schematic structure of SmRho1.1 and SmRho1.2 mutants. Using site-directed mutagenesis, the glutamine Q147 and the valine V148 of SmRho1.1 were substituted by a glutamic acid and a methionine (SmRho1.1 EM) and then the lysine K151 and the serine S153 by two asparagines (SmRho1.1 EMNN). SmRho1.2 QV and SmRho1.2 QVKS mutants were produced by site- directed mutagenesis using SmRho1.2 protein. First, the glutamic acid E147 and

the methionine M148 were substituted by a glutamine and a valine and then, the two asparagines N151 and N153 were replaced by a lysine and a serine. (B) Co-immunoprecipitation and WB experiments performed in *X. laevis* oocytes revealed that SmRho1.2 mutants (HA-tagged) are not able to bind SmHDAC8 (Myc-tagged). cRNAs encoding HA-tagged SmRho1 isoforms, SmRho1.1 mutants or SmRho1.2 mutants were co-injected in *X. laevis* oocytes with cRNA encoding Myc-tagged SmHDAC8. Oocytes were incubated in ND96 medium and lysed. Proteins from soluble extracts were immunoprecipitated (IP) by anti-HA or anti-Myc antibodies and analyzed by WB to detect SmHDAC8 (50 kDa), SmRho1 isoforms (22 kDa) or SmRho1 mutants (22 kDa) with anti-Myc or anti-HA antibodies.
(TIF)

**S1 Table. List of primers.**
(XLSX)

**S2 Table. List of proteins and Uniprot numbers used in phylogenetic analysis.**
(XLSX)

**S3 Table. SmRho1 partners from Co-IP/MS analysis.** Sheet "IP1-IP2 full list" contains the 1,672 different proteins identified from the two independent Co-IP/MS experiments IP1, IP2 respectively. Sheets "IP1 SmRho1-selected protein" and "IP2 SmRho1selected proteins" contain the 86 and 32 different proteins obtained after the selection step (cf. manuscript for details) for IP1 and IP2 respectively.
(XLSX)

## Acknowledgments

The authors wish to thank Jacques Trolet for his technical support, Pr. Christoph Grevelding for providing the SmDia-RDB construct and Dr. Christophe ROMIER for providing the pnEA-tH bacterial expression plasmid.

## Author Contributions

**Conceptualization:** Lucile Pagliazzo, Stéphanie Caby, Raymond J. Pierce.

**Funding acquisition:** Raymond J. Pierce.

**Investigation:** Lucile Pagliazzo, Stéphanie Caby, Julien Lancelot, Sophie Salomé-Desnoulez, Jean-Michel Saliou, Tino Heimburg, Thierry Chassat, Katia Cailliau.

**Methodology:** Jean-Michel Saliou, Thierry Chassat, Katia Cailliau.

**Project administration:** Lucile Pagliazzo, Stéphanie Caby, Jérôme Vicogne, Raymond J. Pierce.

**Resources:** Lucile Pagliazzo, Stéphanie Caby, Julien Lancelot, Sophie Salomé-Desnoulez, Jean-Michel Saliou, Thierry Chassat, Katia Cailliau, Wolfgang Sippl, Jérôme Vicogne.

**Supervision:** Wolfgang Sippl, Raymond J. Pierce.

**Validation:** Sophie Salomé-Desnoulez, Jean-Michel Saliou, Tino Heimburg, Thierry Chassat, Katia Cailliau, Wolfgang Sippl, Jérôme Vicogne, Raymond J. Pierce.

**Visualization:** Lucile Pagliazzo, Stéphanie Caby, Julien Lancelot.

**Writing – original draft:** Lucile Pagliazzo.

**Writing – review & editing:** Lucile Pagliazzo, Stéphanie Caby, Julien Lancelot, Sophie Salomé-Desnoulez, Jean-Michel Saliou, Katia Cailliau, Wolfgang Sippl, Jérôme Vicogne, Raymond J. Pierce.

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
