## [Decision Letter · Decision Letter 0]

22 Jun 2021

Dear Dr. Pierce,

Thank you very much for submitting your manuscript "Histone deacetylase 8 interacts with the GTPase SmRho1 in Schistosoma mansoni" for consideration at PLOS Neglected Tropical Diseases. As with all papers reviewed by the journal, your manuscript was reviewed by members of the editorial board and by several independent reviewers. In light of the reviews (below this email), we would like to invite the resubmission of a significantly-revised version that takes into account the reviewers' comments. 

All reviewers were generally positive about the manuscript. However, several major points were raised which should be addressed in a revised version of the manuscript. The question of the specificity of the antibody against SmRho1 used for immunoprecipitation experiments is especially important.

Regarding the organization of the manuscript, two of the reviewers found the combination of results and discussion confusing, and we agree that separating the extensive results from their possible implications and comparisons to other models will improve the clarity of the manuscript. The reviewers also made many other relevant editorial suggestions and identified many details that need to be clarified in the text.

We cannot make any decision about publication until we have seen the revised manuscript and your response to the reviewers' comments. Your revised manuscript is also likely to be sent to reviewers for further evaluation.

Sincerely,

Uriel Koziol

Associate Editor

Simone Haeberlein

Deputy Editor

Reviewer's Responses to Questions

**Key Review Criteria Required for Acceptance?**

**Methods**

-Are the objectives of the study clearly articulated with a clear testable hypothesis stated?

-Is the study design appropriate to address the stated objectives?

-Is the population clearly described and appropriate for the hypothesis being tested?

-Is the sample size sufficient to ensure adequate power to address the hypothesis being tested?

-Were correct statistical analysis used to support conclusions?

-Are there concerns about ethical or regulatory requirements being met?

Reviewer #1: The methods and clearly and completely described. No concerns.

Reviewer #2: -Are the objectives of the study clearly articulated with a clear testable hypothesis stated?

Not completely

-Is the study design appropriate to address the stated objectives?

Not completely (see protein purification problem)

-Is the population clearly described and appropriate for the hypothesis being tested?

-

-Is the sample size sufficient to ensure adequate power to address the hypothesis being tested?

-

-Were correct statistical analysis used to support conclusions?

- 

-Are there concerns about ethical or regulatory requirements being met?

No

Reviewer #3: The objectives are clear. Some parts of the methods are unclear, see further below.

**Results**

-Does the analysis presented match the analysis plan?

-Are the results clearly and completely presented?

-Are the figures (Tables, Images) of sufficient quality for clarity?

Reviewer #1: Analysis of results matches the plan. The results are clearly and completed presented. The parasite images are striking. Consider making Fig. 6 into two figures?

Reviewer #2: -Does the analysis presented match the analysis plan?

Yes

-Are the results clearly and completely presented?

Not completely.

-Are the figures (Tables, Images) of sufficient quality for clarity?

Yes

Reviewer #3: Overall the results are clear, but at parts further clarification is needed.

Figure legends should be improved.

(see comments below)

**Conclusions**

-Are the conclusions supported by the data presented?

-Are the limitations of analysis clearly described?

-Do the authors discuss how these data can be helpful to advance our understanding of the topic under study?

-Is public health relevance addressed?

Reviewer #1: Succinct and complete

Reviewer #2: -Are the conclusions supported by the data presented?

Not completely.

-Are the limitations of analysis clearly described?

Not completely, again, see protein purification problem.

-Do the authors discuss how these data can be helpful to advance our understanding of the topic under study?

Yes

-Is public health relevance addressed?

Yes

Reviewer #3: The conclusions are valid. Public health relevance is not addressed.

**Editorial and Data Presentation Modifications?**

Reviewer #1: Line 62: There are indeed 200 million individuals infected with schistosome worms, but not sure if 200 million are infected with S. mansoni.

Line 626: Typo

Line706: Typo (its)

Line 728: ‘… lysine K136 …’ seems redundant. Suggest either K136 or lysine 136. This occurs elsewhere, e.g., lines 779 – 781.

Reviewer #2: (No Response)

Reviewer #3: Overall corrections suggested for improving the manuscript:

- “Schistosoma mansoni” does not have to be written out every time, but rather after its first appearance, use “S. mansoni” only.

- Same also for “Xenopus laevis” that should be changed to “X. laevis” after its first appearance.

- Make sure species names are all in italics (e.g. line 163, etc)

- Consistently use units: e.g. write mg/mL or mg mL-1 consistently (and not mg. mL-1)

- If possible, all units of concentration should be given in molarity (especially for the treatment, do not mix mg/mL with uM)

- Consistently use a space between number and unit (e.g. 1 mg/mL, not 1mg/mL, for example line 218)

- Consistently use abbreviations (e.g. “aa”)

- Consistently write “Western blot” (or “western blot”)

- The authors should be consistent with the time forms of verbs and describe the results either all consistently in past, or in present.

- Please indicate for every experiment, how many times it was repeated.

Further minor comments:

Abstract

- Line 29: what is meant by “privileged target”? is it meant “potential target”?

- Line 30: “… Reduction of its transcription…”

- Line 32: comma after “target”

- Line 33/34: “… were identified in the past by…”

- Line 41: introduce the term “immunoprecipitation (IP)” rather here than in line 46

- line 55/56: “… disruption of the schistosomula…”

Author summary

- line 63/64: rewrite more clearly. It should be clearly stated that resistance development is an issue, which might be induced by the MDA programs

- line 64: omit comma after “targets”

- line 66: replace “invalidation” by “reducing”

- line 70: “… novel potential anti-schistosomal drugs. The biological roles…”

- line 71: “… its protein binding partners…”

- line 73: omit comma after “here”

- line 75: omit the word “indeed”

Introduction

- line 85: To the best of my knowledge, PZQ-resistance in schistosomes is a big risk, but there is no proven resistance. Rewrite: “… and risk of resistant parasites.”

- Line 94-95: omit this sentence, as the same information is given three lines further down

- Line 96: “… reduction of SmHDAC8 transcripts…”

- Reference for line 97 missing

- Line 106: “… potential binding partners of SmHDAC8 identified, the…”

- Line 111: omit “through the binding and hydrolysis of GTP”

- Line 121: “… mutation in the budding…”

- Line 126: “… in humans…”

- Line 132: omit “have”

- Line 137: omit comma after “RNAi”

- Line 138: omit comma after “organization”

Materials and Methods

- Line 154: omit dot at end

- Line 158: “before” instead of “previously”

- Line 209: omit dot at end

- Line 234/244: this sentence can be omitted, as it is absolute basics

- Line 235: full stop after Coomassie Blue. Five bands…

- Line 278: rotator at which speed

- Line 317: “in vitro” in italics

- Line 320: “Prof. C. Grevelding…”

- Line 337: “respectively” after “tags, “

- Line 347: source of antibodies missing

- Line 350-368: is there a reference available for the RNAi method applied?

- Line 383: “Parasites treated with corresponding amounts of Dimethyl…”

- Line 397: version of FIJI missing

Results and Discussion

- Line 415: instead of “They” write “SmRho1.1 and SmRho1.2”

- The sentence from line 417-420 should be moved before the sentence in line 415.

- Line 422: “CAAX motif (…”

- Line 425: check wording “anticipated”. What is meant here?

- Line 433: “aa” instead of “amino acids”

- Line 440: “… metazoans, we constructed…”

- Line 442: “from vertebrates, insects, nematodes, cestodes, trematodes and ascomycetes…”

- Line 583: just “Co-IP”, no need to give here again the full term, same for “MS”

- Line 586: “… pre-immune sera…”

- Line 589: “… proteins of each…”

- Line 599: no comma before “such”

- Line 601: “… as a SmRho1 partner…”

- Line 614: “… were identified in Co-IP with SmRho1”

- Line 618: “… SmRho1 interactome in our experiments”

- Line 620: Reference missing

- Line 652: “… SmHDAC8 as SmRho1-interacting proteins”

- Line 656: “… masked the…”

- Line 659: “… to be kept in mind…”

- Line 660: delete “in fact”

- Line 661: Reference missing

- Line 663: “in vitro” in italics

- Line 667: “… all the known protein partners of human RhoA…”

- Line 754/755: this sentence has to be rewritten. It is not clear, what the authors want to state here.

- Line 769: “… and showed…”

- Line 786: “… SmRho1.2 were incapable…”

- Line 845: “… specifically to SmDia”

- Line 846: “As we have shown a specific…”

- Line 848: “… and the Rho Binding…”

- Line 849: “… using the Y2H…”

- Line 850: “In Fig7A we show…”

- Line 859: “… requires further investigation.”

Conclusion

- Line 1064/1065: “… and their partners, it should be kept in mind…”

- Line 1069: “… allow to obtain…”

Figure 1

- Legend very well and clearly written

- Remove lines around subfigures

- Line 513: “… X = terminal AA)”

- Line 516: “human”

- Line 518: “semi-conserved”

- Line 521: “72% and 73% respectively”

- Line 525: could the 310-helices be shown in another color than cyan to distinguish better from alpha-helices?

- Line 691: only “Co-IP” not full wording needed

- Line 733: “… expression systems (Fig. 6).”

- Line 734: “right panel”

- Line 735: “… SmRho 1.1 could grow…”

- Line 740: “… confirm the specific…”

- Line 742: “… it will be necessary…”

- Line 745: “MS” instead of “mass spectrometry”

- Figure 10 is stated first in line 866, before Figure 8. Thus the numbering of the figures should be chronologically adapted.

- Line 902: “… of the parasite…”

- Line 911: “… We could, however, observe that TH65 seemed to…”

- Line 915: “… TSA were highly…”

- Line 918: “… revealed…”

- Line 923: “… on the actin cytoskeleton…”

- Line 924/925: “… that the effects of TH65 could be due to…”

- Line 931: “… organization, but in schistosomula we observed…”

- Line 938: comma after “enzyme”

- Line 939: “… we can nevertheless not…”

- Line 945: “… we will also…”

Figure 2: 

- As the figure contains information on Rho1 of many different species, it would be helpful for the reader, if you would highlight the Schistosome proteins (for example with *).

- Line 578: “in red rectangle”

Figure 3:

- It would be preferred, if the text was bigger, and the piechart smaller

- In the legend, the information on the % given is missing

Figure 4:

- Line 716/717: “… were immunoprecipitated respectively…”

- Line 720: state which IgG mouse antibody was used for the MOCK experiment (also not given in the Methods section)

- Legend is incomplete. How much protein was loaded? Clearly indicate, which ones are the eluates.

- How many times were the tests repeated? Please indicate.

Figure 5:

- line 729: “… spectrum of SmRho1…”

- legend incomplete for this figure. What do the colors mean?

- How many times were the tests repeated? Please indicate.

Figure 6: 

- legend incomplete. Methods and results are repeated here, but no clear description of the figure is given.

- Subfigure a: line 818(819: “… medium (left panel) and diploids… medium (right panel).”

- In subfigure B, indicate “SmHDAC8-Myc” and “SmRho1-HA”, instead of only “Myc” and “HA”

- Subfigure C: mutants do not appear in same order in figure and legend, please correct

- Subfigure C: line 835: “given in red…”

- Why is this figure given in two parts with two titles? Maybe the figure should then rather be given as two separate figures, if sub-titles are needed

- How many times were the tests repeated? Please indicate.

Figure 7: 

- legend incomplete. Methods and results are repeated here, but no clear description of the figure is given.

- Subfigure B, line 890: “Co-IPA”

- In subfigure B, indicate “SmHDAC8-Myc” and “SmRho1-HA”, instead of only “Myc” and “HA”

- How many times were the tests repeated? Please indicate.

Figure 8/9:

- Indicate the concentrations of inhibitors also in the Figure with the respective inhibitors. E.g. “TH65 (50 uM)”

- It is unclear, which negative control was used for what (DMSO/water)

- Give all the inhbitors in uM.

- Clearly state the names of the RNAi sondes also in the figure, as stated in the figure (dsLuc, dsSMHDAC8, dsSmRho1)

- “… Actin-F was stained with phalloidin (green)”… “DAPI (blue)”.

- How many times were the stainings performed? How many times was the qPCR prerformed? Please indicate. Also, if means and SDs are given in subfigure B.

- Line 1036: “… bar represents…”

Figure 10

- Very nice overview, but the figure is very incomplete. Much more information on the figure should be given here.

- Line 1084: comma after “SmRho1.1”

**Summary and General Comments**

Reviewer #1: The manuscript by Pierce and co-workers examines the mechanism of killing after inhibition of HDAC 8 in Schistosoma mansoni. HDAC8 is a validated and druggable target for schistosomiasis drug development. The study is timely, important, well presented, and well written. Linking the activity of HDAC inhibition to small GTPase Rho1 are clear and compelling. The manuscript is very long and presents a very large amount of data; it might be easier to handle as two stories, but the studies are complimentary and could be published as is. 

Disruption of actin was more apparent in schistosomula than adults, but HDAC inhibitors are active against both worm stages. How is this interpreted?

Reviewer #2: The paper of Pagliazzo et al. deals with the further characterization of Histone deacetylase 8 (SmHDAC8) and its potential interaction partners in S. mansoni. This an important topic of schistosome research in the context of basic biology but also towards the identification of potential targets and the development of new drugs.

The Pierce group has much expertise in this field, and here they describe the molecular and cellular characterization of two isoforms of SmRho1, SmRho1.1 and SmRho1.2 as potential SmHDAC8 partners. Among others, the authors found evidence for a role of SmHDAC8 in cytoskeleton organization, a process in which SmRho1.1 may be involved.

This is an interesting study with a number of novel results that will be of interest to readers of PLoS NTD. Furthermore, the results extent existing knowledge about Rho signaling-pathways in schistosomes. However, before the manuscript can be considered for publication the authors must address the following major and minor points.

Major comments:

Lines 192 ff: protein purification worked to some extent. However, the purified protein appears not to be “clean”; there are additional proteins co-purified that have been subjected to mice as well for antibody production. Here the statement of “monospecificity” (line 213) seems not to be justified.

The authors should comment on this, and they should discuss in the paper whether this impurity might have influenced their results such as the Co-IP experiments and the subsequent MS analysis.

Line 606 ff: the authors argue about a parasite protein complex consisting of SmRho1, SmROCK1, SmLIMK, and the Myosin light chain. If it exists, one would expect that these partners share the same compartments/organs for expression. Have the authors searched in the SchistoCyte Atlas (http://www.collinslab.org/schistocyte/) and/or other sub-transcriptome resources to find additional support for their assumption? If not, this has to be done, and the results reported.

Generally, the results are filled with hints to others studies when molecules and their potential functions are reported. Much of this is discussion/conclusion content and should be remove from the results section, which should focus on the obtained data. Furthermore, there are many “ifs and buts” in the results section, which should be removed or rephrased to avoid losing trust and credibility in this study.

This also applies to the conclusion in lines 665-666, in which the authors claim that their (so far presented) results show SmRho1 involvement in cytosolic processes regulating cytoskeleton function, as does the human ortholog, HsRhoA. This conclusion requires deeper analysis that is much more functional, and it is not justified based on the interaction partner analysis, not only because the analysis was based on partners identified by Co-IP with an antibody whose purity was low.

The authors should tune down their functional claims here and elsewhere, as they do not have sufficient experimental support for this. Supporting evidence for SmHDAC8- SmRho1.1 interaction (and SmRho1.2-SmDia, a different story) is presented later, and it is restricted to these tandem combinations.

Line 905: TSA inhibits class I and II mammalian histone deacetylases. How many of these deacetylases (paralogs) exist in S. mansoni? The authors should comment in their manuscript about additional potential inhibitor targets in S. mansoni. In case more than one HDAC paralog exist, have the authors checked and confirmed that the dsRNA part used is specific for their gene? And how do these phenotypes compare to other studies using HDAC8

inhibitors in S. mansoni, such as the one from Saccocia et al. 2020 (PMID: 31661956), which the authors fail to cite?

Line 935-935: With 22%, knock-down efficiency in adults, silencing of SmHDAC8 cannot be considered as robust. How efficient was silencing efficiency of rho1 in somules and adults, and did dsRNA discriminate between the isoforms?

Fig. 10: The authors mentioned a previous study about a proven SmDia-SmTK3 (Src-like TK) interaction. How does this fit into the scenario?

Minor comments

Lines 163/325 elsewhere: species names should be given in italics; this applies also to part of the names of restriction enzymes, which originate from species. E.g., in HindIII the first three letters should be in italics. Please check the whole manuscript and correct where needed.

Lines 170-following: which source was used for primer design before RACE was planned, WormBase ParaSite or something else? And which genome version was used, V5 or V7?

Lines 350 ff.: no reference is given for this kind of RNAi approach (one of several); please cite original literature.

Line 378: a Smp identifier should be added to the used reference gene. The citation is not appropriate, the cited paper seems not to be from the schistosome field. Please cite the paper, in which it has been demonstrated that PSMB7 is a suitable reference gene for S. mansoni.

Lines 381/383: … 4/2 microgram or 4/2 micromolar? Please, harmonize units.

Line 411: provide the Smp identifiers for Rho1.1 and 1.2.

Line 448: to support their assumption of the phylogenetic analysis, the authors should include the orthologs of S. rodhaini, the closest relative of S. mansoni (see Lawton et al. 2011, PMID: 21736723).

Line 601: add Smp identifier for cofilin (and also for all other S. mansoni gene/proteins reported about).

Line 612: “… once again”? May be I missed it, but where have the authors illustrated the involvement of the GTPase in the regulation of cytoskeletal events before?

Line 661: is there a reference for the statement of low SmHDAC8 expression in adults?

Lines 911-912: correct grammar “We can … observed” … .

Better: We have observed, or simply: TH65 seemed to impact …

References: missing italics with species names

Reviewer #3: Pagliazzo and colleagues here report on an impressive study on the characterization of the Schistosoma mansoni protein SmHDAC8 and in particular its interaction partners, mainly SmRho1. They hereby build on a previous study, where interaction of SmHDAC8 and SmRho1 was already identified. The authors here performed further analyses of the SmRho1 isoforms SmRho1.1 and SmRho1.2 on adult schistosomes and schistosomulae and show by Y2H experiments, and expression in Xenopus oocytes, that SmRho1.1 (not SmRho1.2) directly interacts with SmHDAC8. They follow-up on this with several SmRho1 mutants and find that the C-terminal part of SmRho1.1 is important for the interaction. Further, the authors show via Y2H mating experiments, and expression in Xenopus oocytes that SmRho1.2 interacts with SmDia. By immunofluorescence of parasites where SmHDAC8 or Rho1 were inhibited or knocked-down, they show that SMHDAC8 and SmRho1 is involved in cytoskeletal organization, in particular in schistosomulae.

Overall, the study is quite well written and follows a clear structure. The authors chose to merge results and discussion, which makes it difficult, at parts, to understand what was clearly performed in the present study, and what is rather discussion/speculation or found in other studies. Even though a merge of results and discussion is generally accepted in the PlosNTD guidelines, for the present study, a clear separation of results and discussion would be preferable.

Most parts of the study are clear, but at parts, further clarification is needed. Some suggestions for improvement of the language, to make things more clear, are given further below (minor points). Most of the references are given, but some are missing, as also indicated below.

Major points to be clarified

- The abstract does not include all the major points addressed in the study and should be improved.

- For the Method section, it should be stated what is the name of the here applied S. mansoni isolate (line 155).

- In line 169 it is missing, for how long the oocytes were kept at 19°C.

- Is there a reference available for the provided plasmid from Marek and Romier (line 196)?

- The section from line 226-229 is unclear, as the concentration of the antibody is not given, or at least the ratio of how it was diluted in beads should be stated.

- In the section from line 300-308 the authors should check again the mutant description, it seems that EMNN and QVKS were mixed up.

- The authors note in line 592 that it might be that protein quantities were differing between experiments. What were the protein amounts used for each experiment? Were internal control proteins used for normalization?

- In line 595 the authors write about 86 and 32 proteins that were retrieved in two separate IPs. The authors should (at least in a supplementary table) show this data, and also show, how many and which of the proteins were common in the two setups (e.g. in a Venn diagramme). Following this, in line 599, it is unclear, about which protein set the authors are writing when it is stated “These proteins are involved in…”. Are these all the proteins from the two IPs, or only the shared ones? Thus, this part of the results section needs further clarification.

- The authors backed up their MS data with Western blots with SmHDAC8 and SmRho1 (Fig 4). As they state, SmROCK they did not detect as an interacting protein in their MS data, though expected. Did the authors check the presence of SmROCK in the interactome of SmRho1 by Western blot?

- The authors checked the acetylation of SmRho1 by MS. Why was that not done for SmRho1.1 and SmRho1.2 separately?

- The figure legends are clear in the beginning, but from figure 4 on, the legends appear incomplete.

- From line 739 on, the authors discuss that the specificity of interaction between SmHDAC8 and SmRho1.1 would have to be further confirmed by IP. Why was this not included in this study?

- Why was the efficacy of RNAi so low for schistosomulae (Figure 9)?

- Why were different concentrations of inhibitors used for adult worms and schistosomulae (Figures 8/9)?

Additional comments from editors to be addressed:

- Figure 4A - It seems likely that in the left panel, the labels "IP SmHDAC8" and "IP SmRho1" were incorrectly exchanged

- Figure S3 and S4 - the legends are exchanged as well 

- Figures do not appear in order in the main text (e.g. Fig. 10 appears before Fig. 8).

- Regarding the interaction of SmRho1.2 and SmDia, how do you interpret this result considering that Quack et al 2009 found an interaction by Y2H using (presumably) SmRho1.1, although in mutant activated versions?

- Regarding figure 8, the main text mentions the actin cytoskeleton of the tegument, but the figures also show the actin staining of the subtegumental muscle fibers.

- Regarding figure 9, it is clear that the muscle cells are affected by the treatments - however, at this level of magnification, it is not easy to say what happens regarding the structure of actin filaments (l.920), and the gaps in staining could be for example due to death or deattachment of the muscle cells.

- Regarding figure 9, the orthogonal views presented in B could be included with the projections shown in A.

PLOS authors have the option to publish the peer review history of their article (what does this mean?). If published, this will include your full peer review and any attached files.

Reviewer #1: No

Reviewer #2: No

Reviewer #3: No
---

## [Decision Letter · Decision Letter 1]

24 Sep 2021

Dear Dr. Pierce,

Thank you very much for submitting your manuscript "Histone deacetylase 8 interacts with the GTPase SmRho1 in Schistosoma mansoni" for consideration at PLOS Neglected Tropical Diseases. As with all papers reviewed by the journal, your manuscript was reviewed by members of the editorial board and by several independent reviewers. The reviewers appreciated the attention to an important topic. Based on the reviews, we are likely to accept this manuscript for publication, providing that you modify the manuscript according to the review recommendations. 

Sincerely,

Uriel Koziol

Associate Editor

Simone Haeberlein

Deputy Editor

Both reviewers agree that the changes introduced in the revised manuscript have resolved all of the main issues from the original submission, and were very positive regarding the new version.

Some minor issues remain or have been introduced with the modifications of the manuscript, as detailed by reviewer 3. We also provide a few editorial remarks. We have selected "Minor revision" in order to give you the opportunity to revise these issues before acceptance, as some changes cannot be introduced during proofing. Please review these minor issues in a revised version.

Additional remarks:

- Line 738 - "phalloidin staining revealed higher-order actin structures, forming a three-dimensional actin network." - If we interpreted the figures correctly, this network seems to be the muscle fibers of the subtegumental muscle layer - it would seem clearer to describe it as such.

- Regarding your reply to the reviewers and the associated changes to the text:

"Line 411: provide the Smp identifiers for Rho1.1 and 1.2. -> We have added the Smp identifier for SmRho1.1. For SmRho1.2 no identifier exists yet."

It would make things clearer to the reader if it was explicitly stated that SmRho1.2 is not present among the current gene predictions in Wormbase Parasite, and thus lacks a Smp identifier.

- Line 752 - "inhibition and KO of SmHDAC8" -> KO should be "silencing" or "knockdown"

- Line 819 - "Phosphorylated MLC stimulates the binding of myosin to actin to regulate actin filament assembly" - MLC phosphorylation typically regulates stress fibre formation / actomyosin contractility, but not actin filament assembly. Please revise if necessary.

- Line 845 - "...biological processes regulates by a catenin" -> "...biological processes regulated by a catenin"

- Regarding your reply to our previous suggestion:

" Figure 4A - It seems likely that in the left panel, the labels "IP SmHDAC8" and "IP SmRho1" were incorrectly exchanged ->No, there is no error"

It seemed strange that a very strong band for HDAC8 and a very weak band for Rho1 is seen for IP with anti-Rho1, and viceversa, a very strong band is seen for Rho1 and a very weak band is seen for HDAC8 for IP with anti-HDAC8.

Reviewer's Responses to Questions

**Key Review Criteria Required for Acceptance?**

**Methods**

-Are the objectives of the study clearly articulated with a clear testable hypothesis stated?

-Is the study design appropriate to address the stated objectives?

-Is the population clearly described and appropriate for the hypothesis being tested?

-Is the sample size sufficient to ensure adequate power to address the hypothesis being tested?

-Were correct statistical analysis used to support conclusions?

-Are there concerns about ethical or regulatory requirements being met?

Reviewer #2: all yes

Reviewer #3: (No Response)

**Results**

-Does the analysis presented match the analysis plan?

-Are the results clearly and completely presented?

-Are the figures (Tables, Images) of sufficient quality for clarity?

Reviewer #2: all yes

Reviewer #3: (No Response)

**Conclusions**

-Are the conclusions supported by the data presented?

-Are the limitations of analysis clearly described?

-Do the authors discuss how these data can be helpful to advance our understanding of the topic under study?

-Is public health relevance addressed?

Reviewer #2: all yes

Reviewer #3: (No Response)

**Editorial and Data Presentation Modifications?**

Reviewer #2: (No Response)

Reviewer #3: (No Response)

**Summary and General Comments**

Reviewer #2: The authors made some efforts to improve their manuscript. Although not all of my recommendation have been considered (e.g. see below), this revised version can be accepted for publication. 

From my previous review: 

Line 606 ff: the authors argue about a parasite protein complex consisting

of SmRho1, SmROCK1, SmLIMK, and the Myosin light chain. If it exists, one

would expect that these partners share the same compartments/organs for

expression. Have the authors searched in the SchistoCyte Atlas

(http://www.collinslab.org/schistocyte/) and/or other sub-transcriptome

resources to find additional support for their assumption? If not, this has

to be done, and the results reported.

Answer by the authors: As suggested by the reviewer we have searched the SchistoCyte Atlas.

However, we found no relevant data that allow us to link these proteins in

the parasite. We are open to further suggestions, but for this revision we

have not included our analysis.

Looking deeply into the schistocyte data shows interesting patterns of some of the genes in focus of this study. In some cases, clearly elevated levels of expression can be seen in cells representing muscle and parenchyma. In part, expression levels of some of the mentioned genes appears to be influenced by pairing. Data from this resource would have supported some of the ideas and conclusions of the authors, and may have created additional ideas about their biological functions.

Reviewer #3: The authors have tremendously improved the manuscript, changes were implemented (where it made sense), and if not, convincing reasons were given for not following the suggestions.

The manuscript is much more clear now.

However, there appear still a number of sections of discussion in the results part. These should be clearly separate from the results and thus moved to the discussion.

Discussion sections to be moved

- in paragraph from line 786

- whole paragraphs from line 812, from line 817, and from line 821

- lines 899-904

- lines 949-953

- lines 1032-1036

further minor corrections:

- line 80/81: change to: ... is likely and rendered the development of new therapeutic agents imperative. (or even better: ... likely, and it calls for the development of new therapeutic agents)

- line 105: NTD abbreviation not needed, as not further used throughout manuscript

- in line 481, and many other places, the dot between weight and volume needs to be removed still (for example ug mL instead of ug.mL) (also line 548, 563, 564, and further)

- line 606: state how many technical replicates

- line 723: delete "and" before human, and replace with a comma

- line 877: in the Figure it says SmRho1 for both sides of the venn-diagram. either figure or legend incorrect. please correct.

- Figure 5 is not central, and should be moved to supplementaries

- line 934: change to MS/MS

PLOS authors have the option to publish the peer review history of their article (what does this mean?). If published, this will include your full peer review and any attached files.

Reviewer #2: No

Reviewer #3: No

Figure Files:

Data Requirements:

Reproducibility:

References

---

## [Editor Report · Decision Letter 2]

23 Oct 2021

Dear Dr. Pierce,

We are pleased to inform you that your manuscript 'Histone deacetylase 8 interacts with the GTPase SmRho1 in Schistosoma mansoni' has been provisionally accepted for publication in PLOS Neglected Tropical Diseases.

Best regards,

Uriel Koziol

Associate Editor

Simone Haeberlein

Deputy Editor

---

## [Editor Report · Acceptance letter]

21 Nov 2021

Dear Dr. Pierce,

We are delighted to inform you that your manuscript, "Histone deacetylase 8 interacts with the GTPase SmRho1 in <i>Schistosoma mansoni<i>," has been formally accepted for publication in PLOS Neglected Tropical Diseases.

Best regards,

Shaden Kamhawi

co-Editor-in-Chief

Paul Brindley

co-Editor-in-Chief
